# Cube Kernel: Enabling Local Gradient Flow Across Channels in CNNs for Robust and Efficient Building Segmentation

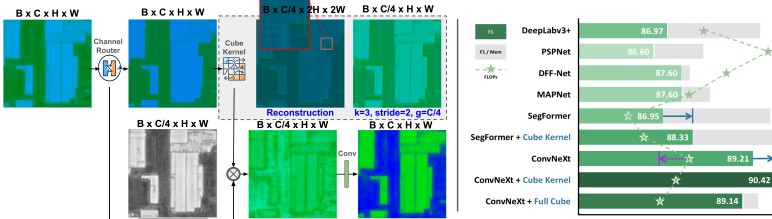

Figure 1: **Overview of the Cube Kernel workflow.** The left shows the internal process: the input features first pass through a Channel Router that adaptively reorders channels, after which per-group spatial attention is computed. The routed features are then reorganized onto a finer spatial lattice through Cube reconstruction and processed by a specially configured convolution for local cross-channel coupling. The output is modulated by the attention map, and a final convolution restores the channel dimension. The right summarizes the resulting accuracy–efficiency improvement.

## Abstract

Understanding inter-band and cross-channel relationships is fundamental to human color perception and object recognition. However, a standard 3×3 convolution kernel provides nine spatial weights and a bias per channel but fuses channel outputs only through a fixed summation. This prevents the operator from learning structured or ratio-like inter-channel cues and limits cross-channel feature coordination. To address this limitation, we develop the Cube Kernel block, a plug-and-play operator that establishes a new computational pathway for local cross-channel coupling. By reconstructing feature channels onto a finer spatial lattice, Cube Kernel enables a single convolution to jointly process and flexibly learn from mixed cross-channel neighborhoods. A learnable Channel Router further adapts channel ordering, while a lightweight spatial attention mask suppresses reconstruction-induced noise. Across CNN-based and Transformer-based backbones, Cube Kernel delivers consistent gains on the WBD, WHU, and Inria datasets. For example, ConvNeXt-U-Cube achieves 90.42% F1 and 82.63% IoU on Inria while reducing parameters and FLOPs by 9.2% and 20.8%, respectively. Ablation studies isolate the contributions of reconstruction, routing, and attention, and gradient analyses reveal substantially stronger inter-channel decorrelation. Owing to its lightweight design, architectural compatibility, and ability to be stacked across layers, Cube Kernel is highly implantable and provides a strong default operator for structured channel mixing in dense prediction tasks.

## 1 Introduction

Urbanization and the push for sustainable cities have heightened the need for precise urban feature detection, especially building segmentation (Fadhel et al., 2024; Schrage, 2024). Accurate building segmentation is a core enabler for resource management and urban planning (Bittencourt et al., 2024). A large body of work has applied deep neural networks to building extraction and has reported strong results (He et al., 2022; Ji et al., 2019; Zhang et al., 2019). However, conventional CNN-based pipelines often struggle to meet the required precision under tight compute budgets, which in turn hinders automation of downstream tasks such as vectorization, cartographic production, and GIS-based statistical analyses.

Standard convolutional kernels operate with fixed channel specificity, which limits their capacity to adaptively integrate information across channels (Lin et al., 2017). This design often yields

redundant or highly correlated channel outputs (Bonet Solé et al., 2022), reducing feature diversity and wasting parameters. Although skip connections (Targ et al., 2016; Wu et al., 2019; Koonce & Koonce, 2021), atrous spatial pyramid pooling (ASPP) (Lian et al., 2021; He et al., 2019), and multi-scale branches (Szegedy et al., 2016; 2017) enhance spatial representation, they largely rely on indirect mechanisms rather than directly learning local cross-channel relationships.

However, because the underlying convolutional mechanism remains unchanged, these designs still lack truly adaptive cross-channel feature extraction and fusion (Schrage, 2024; He et al., 2024; Bibri, 2023; Son et al., 2023). In a typical convolution layer, kernels learn features by optimizing their parameters through backpropagation (Cui et al., 2017), enabling flexible receptive fields and diverse representations (Mairal et al., 2014). We observe that their fixed channel specificity restricts the integration of cross-channel information (Lin et al., 2017). Prior studies report that this often yields redundant or near-duplicate channel outputs: for example, (Li et al., 2023) introduce split–transform–fuse strategies to curb redundancy, while (Bonet Solé et al., 2022) analyze redundancy via CW-NNK graphs and observe substantial overlaps that vary with depth and regularization. Consistent with these findings, our ablations on WBD further show that even very deep, highly parameterized networks still produce strongly correlated channels.

To overcome these limitations, a series of attention-based architectural innovations has been explored. CBAM (Woo et al., 2018) extends SENet (Hu et al., 2018) with a sequential application of channel and spatial attention. ShuffleNet (Zhang et al., 2017) partitions channels and uses channel-shuffle operations to promote inter-group information exchange. Capsule Networks (Sabour et al., 2017) represent entities with vector capsules and use dynamic routing to model local–global relationships. In parallel, Transformer architectures (Vaswani, 2017) motivated vision models to capture long-range and cross-channel dependencies reflecting a shift toward stronger feature fusion. Despite their success, these methods face persistent challenges in dense prediction: 1.They struggle to recover fine-grained spatial details, lack an intrinsic hierarchical inductive bias for multi-scale context, and incur the high computational cost of self-attention, which limits scalability to the high resolutions common in remote sensing. 2.In segmentation tasks, even Transformer-based approaches often rely on convolutional decoders to reconstruct spatial details, which introduces additional architectural complexity and undermines the notion of a fully end-to-end Transformer solution.

We present Cube Kernel, a lightweight and effective primitive designed to strengthen feature fusion in CNN-based image segmentation. A Cube Kernel block consists of three components: (1) the Cube Kernel, which interleaves channels into a local spatial lattice and applies a specially configured convolution to achieve local cross-channel coupling while preserving spatial structure; (2) a learnable Channel Router that adaptively realigns channels to produce more that adaptively reorders and mixes channels to construct task-relevant groups; and (3) a lightweight Spatial Attention module that is computed on primary features and applied after reconstruction to enhance boundaries and suppress reconstruction-induced noise. These modules enable explicit within-layer cross-channel interaction—without relying on global attention and fully connected layers without additional computation. Our main contributions are as follows:

- **A novel, plug-and-play convolutional operator.** Cube Kernel restructures the channel dimension into local spatial neighborhoods, allowing local gradients to propagate across channels and promoting richer inter-channel cooperation than standard convolution.

- **A parameter-efficient Channel Router.** The router learns near-orthogonal mixing channel that emphasize task-relevant representations and assist Cube Kernel in structured feature aggregation with minimal overhead.

- **Strong accuracy–efficiency trade-offs across backbones.** Integrated into diverse CNN- and Transformer-based architectures, Cube Kernel consistently improves segmentation performance on multiple datasets while reducing parameters and FLOPs.

## 2 RELATED WORK

**Grouped and Cross-channel Convolutions.** Grouped and depthwise convolutions reduce computation by restricting channel interactions, widely adopted in lightweight networks and serving as backbones in segmentation frameworks like DeepLabv3+ (Dong et al., 2020; Cai et al., 2021). While efficient, these operations perform only local or group-level channel mixing, limiting their ability to

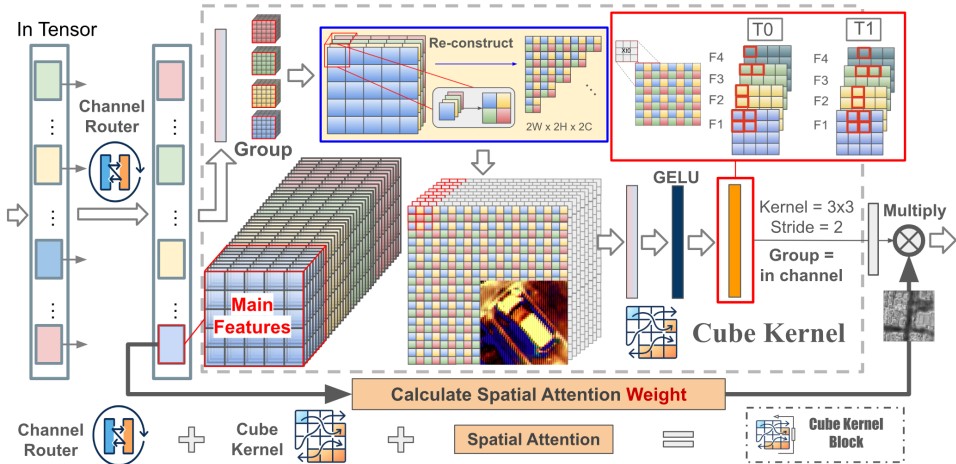

Figure 2: **Overview of the Cube Kernel Block.** The Cube Kernel performs depthwise spatial aggregation on interleaved channels, while the lightweight router facilitates cross-channel flow. A spatial attention computes attention weights to enhance spatially relevant features. This block enables efficient local gradient coupling and structured channel mixing without relying on global attention.

capture complex inter-channel dependencies. To address this, channel-attention mechanisms such as SE-Net (Hu et al., 2018) and CBAM (Woo et al., 2018) recalibrate channel importance, achieving gains tasks but at the cost of losing spatial detail. More recently, involution (Li et al., 2021) introduced location-specific kernels for intra-group feature fusion, demonstrating stronger contextual modeling than standard convolution in tasks requiring fine detail.

**Attention and Transformer-based Models.** Transformers extend cross-channel and long-range spatial reasoning beyond CNNs. The Vision Transformer (ViT) (Dosovitskiy, 2020) models global patch relationships via self-attention but is memory-intensive and less effective at local bias. Variants such as Swin Transformer (Liu et al., 2021), Polygon Transformer (Liang et al., 2020), and Build-Former (Wang et al., 2022) improve efficiency and accuracy in building segmentation by incorporating hierarchical windows or dual-path structures. Hybrid designs like Swin-UNet (Cao et al., 2021) further combine attention with U-Net for multi-scale feature preservation. Despite these advances, existing cross-channel mechanisms either rely on heavy global attention or lightweight reweighting, leaving a gap for approaches that achieve efficient, structured, and spatially-aware channel mixing.

**Location-specific kernels.** Involution, proposed by Li et al. (2021), generates location-specific kernels conditioned on spatial positions, thereby enhancing adaptability to spatial variations. However, involution still extracts features from each pixel's channels independently, limiting its ability to capture complex inter-channel relationships (He et al., 2024). Depthwise separable convolution (Howard et al., 2017) factorizes standard convolution into depthwise and pointwise steps, greatly reducing computation, but similarly treats channels in isolation before final mixing. Other variants, such as dilated convolution (Yu & Koltun, 2016), deformable convolution (Dai et al., 2017), and dynamic convolution (Chen et al., 2020), improve spatial sampling flexibility or adapt kernel weights dynamically, yet they also rely on indirect mechanisms for channel integration.

None of these approaches explicitly enforce local cross-channel gradient coupling at the convolutional level, as most rely on spatial adaptivity or efficiency improvements, while cross-channel interactions are typically handled indirectly through global pooling or attention.Cube Kernel addresses that. Cube Kernel directly addresses this gap by enabling group-based local cross-channel cooperation through spatial interleaving, and flexible channel reordering.

## 3 CUBE KERNEL

In a standard $3 \times 3$ convolution, each output channel is produced by applying a $C_{in} \times 3 \times 3$ kernel to the input, where all input channels are filtered independently and the fused through a fixed channel summation. While the operator can learn rich intra-channel patterns via multiplicative weights $W$ and additive biases $b$, this rigid summation is the only form of cross-channel interaction avail-

able. Consequently, the layer lacks any learnable mechanism for modeling meaningful relationships (structured or proportion-based dependencies) between channels within a local receptive field.

A single linear convolution cannot express multiplicative or ratio-like cross-channel interactions. Although stacking multiple layers may approximate such nonlinear behaviors, it does so inefficient: Multiplication requires multiple layers to merge, and true ratio-based relationships (e.g., division) are extremely difficult to represent with convolution+ReLU compositions. These cues are therefore fundamentally inaccessible to a single convolutional operator. This limitation is particularly relevant for building extraction, where many discriminative cues are inherently ratio-based, including color ratios (e.g., roof vs. pavement), texture contrasts, and proportional shadow–highlight differences.

### 3.1 DEFINITION OF CUBE KERNEL

We propose the **Cube Kernel Block**, a novel operator that enables direct local gradient coupling across channels. As shown in Fig. 2, the Cube Kernel Block includes **A. Cube Kernel**, **B. Learnable Channel Router**, and **C. Spatial Attention**.

**A. Cube Kernel** is a convolutional operator that enables explicit local cross-channel interaction by reorganizing feature channels into shared spatial neighborhoods before spatial filtering. Through grouping and spatial reconstruction, channels that are originally processed in isolation are placed within the same receptive field, allowing each kernel weight to simultaneously interact with multiple channels. This structural introduces a learnable cross-channel pathway that standard convolution fundamentally lacks. Given $\mathbf{X} \in \mathbb{R}^{H \times W \times C}$, Cube Kernel proceeds in two steps:

(1) Grouping and Reconstruction. We partition the channels into 4 groups of channels $C/4$. For each index $k \in \{1, \ldots, C/4\}$, the $k$-th channel from all four groups is placed into the four spatial positions of a $2 \times 2$ lattice. This produces a deterministic reconstruction:

$$\mathbf{X} \in \mathbb{R}^{H \times W \times C} \xrightarrow{\mathcal{R}} \mathbb{R}^{2H \times 2W \times (C/4)} \xrightarrow{\text{Conv}_{3 \times 3, s=2, g=C/4}} \mathbf{Y} \in \mathbb{R}^{H \times W \times (C/4)}, \tag{1}$$

where $\mathcal{R}$ performs the fixed $2 \times 2$ spatial re-indexing (reconstruction), interleaving channels onto a finer lattice.

(2) Specially Configured Convolution. The interleaved tensor is then processed by a $3 \times 3$ grouped convolution (groups $= C/4$). By applying the grouped $3 \times 3$ convolution on the reconstructed feature map, receptive field extends across both the spatial neighborhood and the regrouped channels. Each filter operates exclusively on its corresponding reconstructed group, and stride $\sim 2$ restores the original spatial resolution.

This decoupled design aligns with the evolution of modern architectures: MobileNets separate pointwise and depthwise operations; ConvNeXt integrates Transformer-style channel mixing; Vision Transformers and MLP-Mixers fully decouple spatial and channel processing. Cube Kernel follows this principle while preserving the convolutional inductive bias, providing an explicit and learnable pathway for local cross-channel coordination that standard convolution fundamentally lacks.

**B. Learnable Channel Router** is positioned before the Cube Kernel's spatial mixing. Cube Kernel interleaves features onto a finer spatial lattice, but the fixed grouping constrains channel selection, which may be sub-optimal for decoding. Therefore, we introduce a learnable Channel Router, a lightweight $1 \times 1$ convolution that performs a per-location, channel-only linear map $\mathbf{Y}(h, w, :) = \mathbf{W}_r \mathbf{X}(h, w, :) + \mathbf{b}_r$, making the block transparent at the beginning. Because the transform preserves spatial indices, training drives $\mathbf{W}_r$ toward near-orthogonal mixes, effectively realizing soft channel reconstruction that preconditions the subsequent interleaving and stride-2 depthwise aggregation. The router also introduces explicit cross-channel coupling in backpropagation with $\boldsymbol{\delta}(h, w, :) = \frac{\partial L}{\partial \mathbf{Y}(h,w,:)}$, the gradient $\frac{\partial L}{\partial \mathbf{W}_r} = \sum_h^H \sum_w^W \boldsymbol{\delta}(h, w, :)\mathbf{X}(h, w, :)^\top$ aggregates signals across channels while leaving the spatial lattice unchanged, yielding adaptive emphasis of task-relevant channels at negligible overhead.

To assess how initialization shapes the Learnable Channel Router, we integrate Cube Kernel into DeepLabV3+ and visualize the full 304×304 router weight matrices (Left of Fig. 3).

Across training, the router's weight maps remain diagonal-dominant; under identity initialization the bright diagonal smooths into a near-orthogonal mixing. This is an emergent effect of grouping



Figure 3: **Left:** Router weight matrices evolution under different initialization. **Right:** Gradient-similarity heatmaps (Conv vs. Router). The router shows richer color and faint off-diagonal streaks near C=76, 152, 228.

Figure 4: **Standard convolution backpropagation.** Pseudo-code illustrating channel-wise gradient flow and fixed additive cross-channel fusion.

and optimization (not a hard constraint). Its persistence indicates the model actively selects identity-biased yet flexible mixing that reduces inter-group redundancy and supports spatial–channel reconstruction. In the right panel of Fig. 3, the gradient–similarity maps show a clear contrast: a plain $1\times1$ convolution yields values clustered near zero (noise-like, decorrelated, and weakly structured), whereas the Cube Kernel router exhibits coherent red/blue structure with faint off-diagonal streaks near the group boundaries ($\approx C/4, C/2, , 3C/4$). For $C=304$ (four groups of 76), these bands align with $76, , 152, , 228$, indicating *functional clusters*: higher within-group gradient coherence while retaining intra-group diversity, consistent with the reconstruction design (see Fig. 5 and App. C). Together, these observations suggest that the router's optimization encourages greater inter-group separability while reducing intra-group conflict, enabling channel reassignment and cooperative feature aggregation—key contributors to Cube Kernel effectiveness.

A more detailed analysis of how Cube Kernel reshapes the backward gradient flow and enables cross-channel coupling within a single receptive field is provided in Appendix C.

**C. Spatial Attention** is computed from the main feature branch before Cube Kernel reconstruction. For each location, average- and max-pooled responses along the channel axis are concatenated and passed through a $7 \times 7$ convolution and sigmoid to produce an attention mask $A \in [0,1]^{H \times W}$. Unlike immediate modulation, $A$ is delayed until after Cube Kernel reconstruction, yielding $\mathbf{Y}(h,w,:) = \hat{\mathbf{F}}(h,w,:) \odot A(h,w)$. This deferred gating emphasizes rooftop boundaries while suppressing reconstruction noise, stabilizing training, and allowing Cube Kernel to function as a stackable, reusable operator.

## 3.2 Benefits from Cube Kernel

**New pathway for forward and backward interaction.** Cube Kernel removes the structural constraint of standard convolution by constructing a joint spatial–channel processing pathway. In a standard $3\times3$ convolution, each input channel is filtered independently and then fused only through fixed additive summation, restricting cross-channel interaction to a purely additive form. This produces channel-independent (separable) gradients, preventing the layer from learning proportion-based or multiplicative relationships between channels.

Spatial reconstruction changes this behavior fundamentally. By interleaving channels into the same local neighborhood, multiple bands contribute jointly to the same kernel computation. Although the convolution operator itself remains linear, the *effective* forward relation among channels becomes jointly aggregated rather than independently added.

This structural design fundamentally reshapes the backward gradient flow. In conventional convolutions, the gradient of the loss function $L$ with respect to the weight matrix $W$ follows a two-step chain rule: the global gradient $\frac{\partial L}{\partial Z_i}$, which accumulates loss signals at the output, and the local gradient $\frac{\partial Z_i}{\partial W_{i,j}}$, which captures how each weight affects the layer output activation:

$$\frac{\partial L}{\partial W} = \sum_{i=1}^{C_{\text{out}}} \sum_{j=1}^{C_{\text{in}}} \frac{\partial L}{\partial Z_i} \cdot \underbrace{\frac{\partial Z_i}{\partial W_{i,j}}}_{\text{Local Gradient}} \rightarrow \frac{\partial L}{\partial W_{\text{Cube}}} = \sum_{i=1}^{C_{\text{out}}} \sum_{j \in G} \frac{\partial L}{\partial Z_i} \cdot \underbrace{\frac{\partial Z_i}{\partial W_{\text{Cube},i,j}}}_{\text{Local Gradient (Cross Channel)}} \quad (2)$$

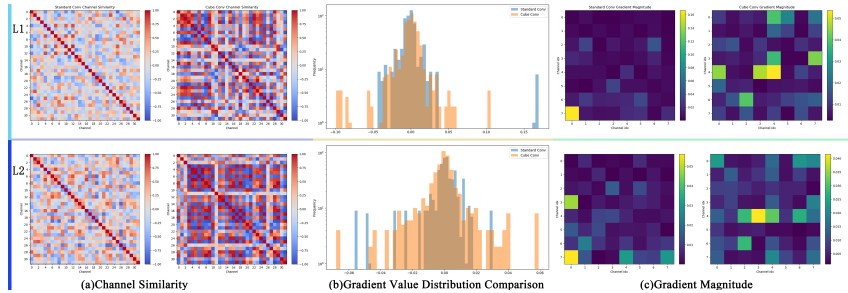

Figure 5: **Visualizations of kernel gradient comparison between standard conv. kernel and Cube Kernel(Local).** (a) channel-wise cosine-similarity matrices, (b) gradient-value histograms, and (c) gradient-magnitude heatmaps.

Unlike traditional settings where local gradients are isolated within individual channels, the Cube Kernel introduces cross-channel interactions directly into the local gradient path. Through grouped reconstructions and spatial interleaving, each gradient update $\frac{\partial Z_i}{\partial W_{\text{Cube},i,j}}$ is influenced by multiple channels within the same group $G$. This mechanism enables the kernel to capture joint channel dependencies, allowing the network to identify which channel combinations are most informative and thereby promoting richer feature fusion. The Fig. 4 illustrates this update process, where the two derivative computations are carried out separately and subsequently combined for weight updates.

**Computational Efficiency.** Cube Kernel offers both efficiency and enhanced representation. At matched output size $H \times W$, a standard $k \times k$ convolution costs Params is $k^2 C_{in} C_{out}$ and MACs is $HW k^2 C_{in} C_{out}$. Our Cube implementation treats interleaving as index-only ($\approx 0$ cost) and comprises three lightweight pieces: (1) a Router $1 \times 1$ over $C$ channels with grouping $G$, (2) a depthwise $3 \times 3$ executed at $2H \times 2W$ with stride-2, and (3) an output fusion with kernel $k$ and grouping $G$. Therefore, Cube Kernel totals $13,456$ parameters and $2.21 \times 10^8$ MACs, versus $36,864$ parameters and $6.04 \times 10^8$ MACs for a standard convolution, delivering earlier channel mixing and stable accuracy gains at substantially lower compute.

**Diverse Gradient Directions.** We probe gradient behavior at the fusion layer of standard convolutional operator and Cube Kernel, and visualize three views for two channels (L1, L2) in Fig. 5: (a) inter-channel cosine-similarity matrices, (b) gradient-value histograms, and (c) gradient-magnitude maps. In (a), the standard conv. kernel shows large red block patterns with near-unity off-diagonals, indicating strong channel co-linearity; by contrast, Cube Kernel breaks this block structure and markedly reduces off-diagonal similarity across both L1 and L2, evidencing decorrelated inter-channel updates. In (b), Cube produces a broader, heavier-tailed gradient distribution with less mass near zero, implying richer and less redundant directions. In (c), Cube yields more spatially distributed high-magnitude responses (brighter cells), reflecting a greater number of active gradient directions. Collectively, these views indicate that Cube Kernel learns more diverse channel features and provides better-conditioned gradients than the standard convolution.

**Preserving Spatial Structure.** Rather than resorting to global attention or additional MLPs, Cube Kernel applies a minimal structural change. We interleave multiple feature channels into a local cube around each pixel, placing values from different channels in adjacent spatial locations. A subsequent depthwise $3 \times 3$ convolution then aggregates these interleaved neighborhoods. Although the convolution itself has no cross-channel weights, interleaving makes features from different source channels co-occur within the same receptive field. As a result, both forward activations and back-propagated gradients naturally exchange information across channels while strictly preserving spatial layout. As shown in Fig. 6, shuffled features preserves the full geometric structure and reveals additional fine details (e.g., sharper roof edges and corners). Compared with global attention, this mechanism achieves similar cross-channel mixing at a small fraction of the memory and compute.

As a result, the backward gradient associated with a Cube Kernel weight is no longer tied to one channel but to a spatial patch containing interleaved features from multiple channels. This establishes a new, explicit pathway for cross-channel coupling in both forward and backward propagation—unlocking joint pattern learning that standard convolution finds difficult to realize explicitly within a single layer and typically only approximates indirectly via deep stacking.

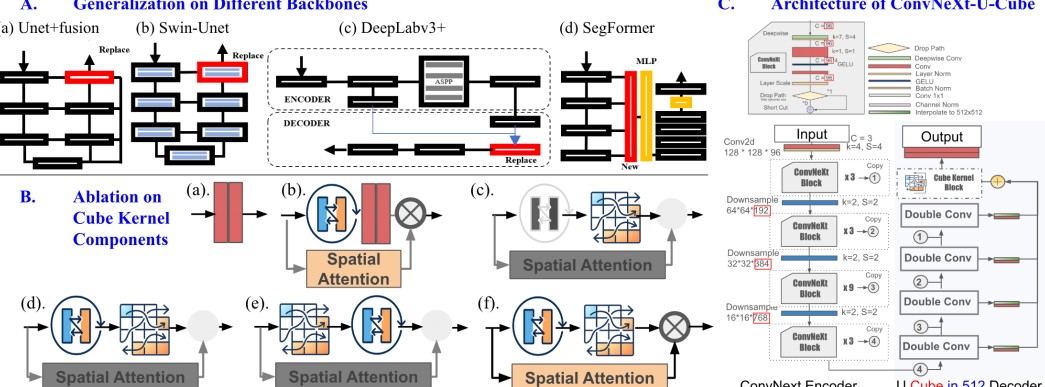

Figure 6: **Examples of Structure-Preserving Channel Interleaving.** Four individual feature maps are interleaved to composite Cube-Kernel feature map, which preserves the global geometry while revealing additional fine details.

Table 1: Performance Comparison to SOTA methods on Inria Aerial Image Labeling Dataset.

| Method | OA | Pre | Rec | F1 | IoU | Params | FLOPs | GPU Mem |
|---|---|---|---|---|---|---|---|---|
| Deeplabv3+(Chen et al., 2018) | 96.85 | 88.03 | 85.67 | 86.97 | 77.30 | 40.00 | 105.00 | 7.20 |
| ASLNet (Ding et al., 2022) | **97.15** | 90.00 | 86.85 | 88.27 | 79.30 | - | - | - |
| PSPNet (Zhao et al., 2017) | - | 88.20 | 85.00 | 86.60 | 76.40 | 65.70 | 262.74 | 12.71 |
| DFF-Net (Chen et al., 2024) | - | 88.80 | 86.30 | 87.60 | 77.90 | 32.15 | 222.49 | 17.85 |
| MAPNet (Tan et al., 2024) | - | 88.90 | 85.70 | 87.60 | 77.40 | 23.85 | 92.26 | 11.61 |
| C_ASegformer (Zhang et al., 2025) | 95.33 | 90.83 | 89.17 | 89.97 | 82.58 | - | - | - |
| SegFormer(Xie et al., 2021) | 96.15 | 93.79 | 81.03 | 86.95 | 76.91 | 13.68 | 15.36 | 7.00 |
| SegFormer + Cube Kernel | 96.49 | 93.06 | 84.07 | 88.33 | 79.11 | 14.55 | 29.72 | 7.00 |
| REIN-DINOv2 (Wei et al., 2024) | 96.93 | 90.84 | 89.62 | 90.23 | 82.20 | 23.57 | - | 19.60 |
| REIN-DINOv2 + Cube Kernel | 97.06 | 90.67 | **90.43** | **90.55** | **82.74** | 23.19 | - | 19.60 |
| ConvNeXt (Liu et al., 2022) | 96.71 | 92.77 | 85.92 | 89.21 | 80.52 | 38.47 | 96.80 | 11.20 |
| ConvNeXt + Cube Kernel | 97.03 | 92.22 | 88.69 | 90.42 | 82.63 | 34.94 | 76.71 | 10.25 |
| ConvNeXt + Full Cube | 96.73 | **93.91** | 84.83 | 89.14 | 80.42 | 29.18 | 44.41 | 9.80 |

Note: OA, IoU, Rec, F1, Pre (%), Params = Parameters (M), FLOPs = Floating Point Operations of $3 \times 512 \times 512$ (G). **Bold** = best, underline = second.

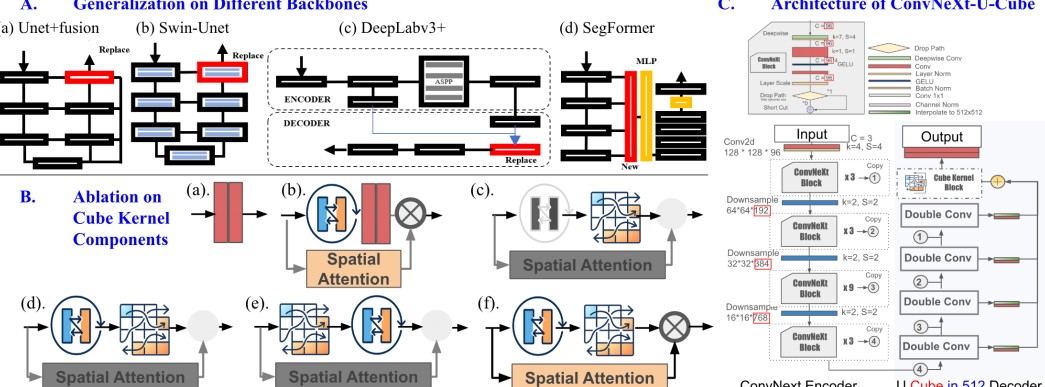

Figure 7: Generalization and Ablation Study Structures

## 4 EXPERIMENTS

We evaluate on three building-extraction benchmarks: (1) WBD (Waterloo Building Dataset) (He et al., 2021), covering the Kitchener–Waterloo region with $100K$ annotated buildings, (2) WHU (Christchurch) (Ji et al., 2019), containing $22K$ buildings and pre-tiled into $512 \times 512$ patches, and (3) Inria (Maggiori et al., 2017), characterized by diverse rooftop morphology and frequent occlusions. All models are trained with Adam and binary cross-entropy (BCE) loss on a single NVIDIA GeForce RTX 3060 (12 GB) with CUDA 12.6. Unless otherwise noted, the initial learning rate is $1 \times 10^{-5}$ with a cosine decay schedule.

### 4.1 COMPARISON TO STATE-OF-THE-ART METHODS

As summarized in Tab. 1, integrating the Cube Kernel into ConvNeXt achieves the best overall results, reaching 97.03% OA, 82.52% IoU, and 90.42% F1, thereby surpassing recent state-of-the-art methods. When incorporated into SegFormer and ConvNeXt backbones, the Cube Kernel delivers an average improvement of about 2% in IoU. This demonstrates that, across both Transformer-based (SegFormer) and CNN-based (ConvNeXt) architectures, Cube Kernel consistently enhances performance, underscoring its flexibility as a plug-and-play module.

Table 2: Performance Comparison on Different Backbones on WBD and WHU Datasets.

| MODEL | WBD (Larger and Complex) | | | | WHU (Smaller and Sample) | | | | Model Complexity | |
|---|---|---|---|---|---|---|---|---|---|---|
| | IoU | Rec | F1 | Pre | IoU | Rec | F1 | Pre | Params | FLOPs |
| UNet (Ronneberger et al., 2015) | 77.75 | 91.04 | 87.48 | 84.19 | 79.38 | 85.02 | 88.56 | 92.28 | 27.22 | 364.00 |
| UNet + Conv Kernel | 79.90 | 92.05 | 89.84 | 87.84 | 82.74 | 86.80 | 89.73 | 92.33 | 27.21 | 402.80 |
| UNet + Cube Kernel | 86.07 | 92.89 | 92.52 | 92.54 | 84.95 | 88.67 | 91.86 | 95.30 | 27.14 | 220.89 |
| DeepLabv3+(Chen et al., 2018) | 85.43 | 88.57 | 92.14 | 96.02 | 86.17 | 89.43 | 92.57 | 95.94 | 41.35 | 178.72 |
| DeepLabv3+ Cube Kernel | 89.03 | 94.68 | 94.19 | 94.01 | 86.93 | 90.69 | 93.01 | 95.46 | 40.00 | 105.00 |
| Swin-UNet (Cao et al., 2021) | 80.89 | 87.05 | 90.02 | 92.95 | 85.00 | 92.69 | 91.89 | 91.10 | 42.03 | 192.70 |
| Swin-UNet + Cube Kernel | 87.26 | 89.87 | 93.60 | 95.29 | 88.59 | 93.95 | 94.72 | 95.21 | 29.80 | 100.58 |
| SegFormer (Xie et al., 2021) | 87.50 | 90.07 | 93.33 | 96.84 | 86.24 | 89.28 | 92.61 | 96.02 | 13.68 | 15.36 |
| SegFormer + Cube Kernel | 88.78 | 92.32 | 94.06 | 95.87 | 91.01 | 93.63 | 95.30 | 97.02 | 14.27 | 25.16 |
| ConvNeXt (Liu et al., 2022) | 84.00 | 87.86 | 91.30 | 95.03 | 90.81 | 91.82 | 95.18 | 98.80 | 38.43 | 94.66 |
| ConvNeXt + Cube Kernel | 89.10 | 92.80 | 94.25 | 96.31 | 91.67 | 94.65 | 95.65 | 96.68 | 34.21 | 72.40 |

Note: IoU, Rec, F1, Pre (%), Params = Parameters (M), FLOPs = Floating Point Operations of $3 \times 512 \times 512$ (G). **Bold** = best, underline = second.

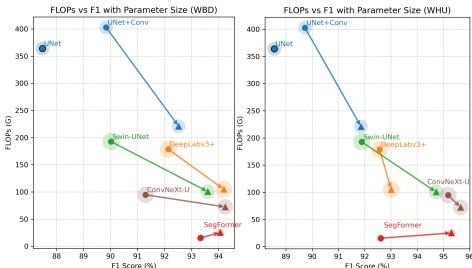

Figure 8: **F1 vs. FLOPs on WHU (left) and WBD (right).** Each colored arrow links the baseline backbone (circle) to its +Cube Kernel variant (triangle). Marker area encodes Params (M).

Table 3: Comparison of Cube Kernel strides on WBD dataset. (%)

| Metric | Normal | Stride=1 | Stride=2 |
|---|---|---|---|
| **IoU** | 79.90 | 83.94 | **86.07** |
| **OA** | 97.98 | 98.47 | **98.67** |
| **Rec** | 92.05 | 92.62 | **92.89** |
| **F1** | 89.84 | 91.27 | **92.52** |
| **Kappa** | 87.61 | 90.43 | **91.78** |
| **Pre** | 87.84 | 89.96 | **92.54** |

On the REIN–DINOv2 pipeline, adding Cube Kernel yields consistent gains with minimal intrusion: F1 : $90.23 \rightarrow 90.55$ (+0.32), mIoU : $82.20 \rightarrow 82.74$ (+0.54); parameters drop from $23.57\,\mathrm{M}$ to $23.19\,\mathrm{M}$, and FLOPs stay at $19.60\,\mathrm{G}$. Because REIN–DINOv2 lacks a conventional decoder, we only replace the lightweight refinement convolutions in the Mask2Former prediction head, leaving the backbone unchanged. These improvements under a constrained insertion point (and on strong DINOv2 features) show that Cube Kernel enables effective cross–channel coordination without deep decoder stacking and has potential utility on the encoder side of foundation–model backbones.

Furthermore, the FullCube variant maintains competitive accuracy while significantly reducing computational cost and memory usage (76.71 G $\rightarrow$ 44.41 G FLOPs), making it a compelling option when efficiency is prioritized. These results highlight not only the robustness of Cube Kernel across different backbone paradigms but also its practical advantages in balancing accuracy and efficiency.

### 4.2 GENERALIZATION ON DIFFERENT BACKBONES

Both the Cube Kernel and the full Cube Kernel Block are evaluated by integrating them into UNet, DeepLabv3+, Swin-UNet, and SegFormer, with the corresponding insertion points illustrated in Fig. 7. For strict comparability, exactly one convolutional layer is replaced in each backbone. The only exception is the MLP-based SegFormer, where an additional layer is introduced, leading to a slight increase in model size.

On two datasets, **ConvNeXt+Cube Kernel** achieves the best results, with F1 scores of 94.25 and 95.65, and IoU scores of 89.10 and 91.67, respectively. Across all four backbones, integrating Cube Kernel consistently improves accuracy, see Tab. 2. The largest gain occurs on UNet, where IoU rises from 77.75% $\rightarrow$ 86.07% (+8.3%). Among all configurations, As shown in Fig. 9, introducing Cube Kernel enables all backbones to recover building footprints more completely, with crisper roof boundaries and crucially far less fragmentation. This indicates that Cube Kernel, by introducing cross-channel coupling at the local-gradient level, effectively strengthens spatial cues and thus complements existing backbones without requiring additional global attention.

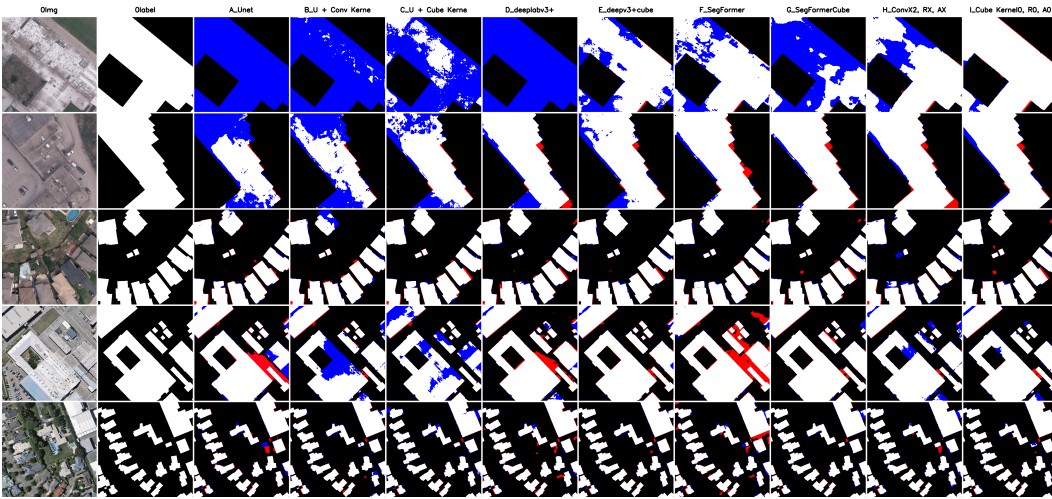

*White: True Positive, Blue: True Negative, Black: False Positive, Red: False Negative.*

Figure 9: Building Detection Examples of Proposed Method and the Comparative Methods.

Table 4: Performance Comparison of Different Components Designs on WBD and WHU Datasets.

| MODEL | WBD (Larger and Complex) | | | | WHU (Smaller and Sample) | | | | Model Complexity | |
|---|---|---|---|---|---|---|---|---|---|---|
| | IoU | Rec | F1 | Pre | IoU | Rec | F1 | Pre | Params | FLOPs |
| DoubleConv, R✗, A✗ | 84.00 | 87.86 | 91.30 | 95.03 | 90.81 | 91.82 | 95.18 | **98.80** | 38.43 | 94.66 |
| DoubleConv, R✓, A✓ | 88.26 | 92.49 | 93.88 | 95.31 | 91.10 | 93.61 | 95.34 | 97.14 | 35.01 | 95.75 |
| Cube Kernel✓, R✗, A✗ | 89.10 | 92.80 | 94.25 | 96.31 | 91.67 | **94.65** | 95.65 | 96.68 | 34.21 | 72.40 |
| Cube Kernel✓, R✓, A✗ | 89.76 | 93.85 | 94.60 | 95.36 | 91.89 | 94.12 | 95.77 | 97.48 | 34.93 | 73.00 |
| Cube Kernel✓, R✓(post), A✗ | 87.88 | 91.43 | 93.54 | **95.76** | 85.65 | 88.97 | 92.86 | 95.80 | 44.39 | 89.81 |
| Cube Kernel✓, R✓, A✓ | **91.15** | 95.07 | **95.37** | 95.68 | **91.92** | 94.03 | **95.79** | **97.62** | 34.94 | 76.71 |
| Cube Kernel×4✓, R✓, A✓ | 91.12 | **95.83** | 95.36 | 94.89 | 91.37 | 93.52 | 95.49 | 97.55 | 31.73 | 51.31 |

Note: IoU, Rec, F1, Pre (%), Params = Parameters (M), FLOPs = Floating Point Operations of $3 \times 512 \times 512$ (G). R = Router, A = Attention. **Bold** = best, underline = second.

From Fig. 8, integrating Cube Kernel improves the accuracy–compute trade-off on both WHU and WBD datasets. The FLOPs of UNet, DeepLabv3+, and Swin-UNet drop from 364G → 221G (0.61 ×), 179G → 105G (0.59 ×), and 192G → 105G (0.55 ×), respectively. Overall, three of the four backbones move down-and-to-the-right in the F1–FLOPs plane (higher F1 with lower compute), underscoring Cube Kernel's ability to boost accuracy while cutting computation.

Additional large-scale experiments on COCO-Instance80 and COCO-Stuff164K further confirm the generality of Cube Kernel across CNN, hierarchical Transformer, and plain-ViT architectures (see Appendix D).

### 4.3 ABLATION STUDIES

**Cube Kernel Components Designs.** To disentangle the contributions of individual components, we run an ablation over three factors: (1) the Cube Kernel itself, (2) the Router, and (3) Attention. Results are reported in Tab. 4. Ablations indicate that Cube Kernel alone surpasses a conventional DoubleConv (IoU 89.10% vs. 84.00%) while being substantially lighter in both parameters and FLOPs. Adding the Router and Attention yield a further gain and provide complementary benefit. These results identify the Cube Kernel as the principal driver of accuracy, with the Router acting as a critical amplifier. Finally, in the FullCube setting, the model attains performance comparable to the best variant while reducing compute from 94.66 G to 51.31 G FLOPs (45.8% ↓), demonstrating strong efficiency without sacrificing accuracy.

**Stride settings.** We further examine the effect of stride in a single Cube Kernel, evaluated on UNet with Cube Kernel fusion at Fig. 7(a). As reported in Tab. 3, Cube Kernel consistently surpasses the standard convolution under all stride settings, with stride=2 yielding the best performance (IoU = 86.07%; +6.17% over standard, +2.13% over stride=1).

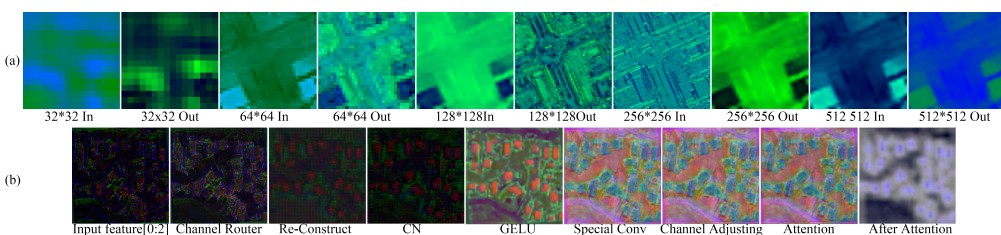

Figure 10: Visualizes the input features for Cube Kernel

## 4.4 DISCUSSION

**Layerwise Feature Evolution and Cube Kernel Internals.** Fig. 10(a) visualizes the first three channels across multiple layers of FullCube. It reveals a clear depthwise hierarchy: at $32 \times 32$, outputs have deviated from inputs, indicating that the Cube extracts coarse textures and global context early. As resolution increases to $128$, $256$, and $512$, the feature maps show strong sensitivity to linear structures such as building and road boundaries. These cues are then propagated to the top layer, reflecting a coordinated progression from global context to fine details for segmentation.

Beyond the qualitative trend, we observe that Cube Kernel avoids dead responses in early layers. For instance, the $128 \times 128$ stage, the captured patterns remain diverse suggesting that important features are progressively routed forward and adapt dynamically according to model requirements. We find that U-Net produced 28 channels that were almost entirely black from 64 channels, whereas Cube Kernel contained only 6 channels. This indicates that Cube Kernel achieves more efficient use of intermediate features, aggregating and transmitting information with a more balanced distribution in feature space.

Fig. 10(b) exposes the internal behavior of the Cube Kernel block. It clearly demonstrates how edge features and the building mask are brought together through the reconstruct group. After convolution, the model successfully extracts building boundaries with enhanced sharpness. Following the Channel Router, building edges become noticeably clearer, indicating selective emphasis and re-assignment of salient signals to more discriminative channels. The reconstruct step further amplifies target regions, while Channel Adjusting and Attention modules suppress spurious responses in non-building areas (e.g., roads). Taken together, the Cube Kernel integrates edge/texture information with building location cues into a unified receptive view, yielding compact, discriminative, and task-aligned high-level representations.

**Looking ahead.** Cube Kernel can be deployed across a wide range of architectures, including CNN-centric and hybrid CNN–Transformer designs, where its robust, structure-preserving channel fusion to improve both accuracy and efficiency. Our codebase is modular and plug-and-play, enabling seamless integration and opening up numerous practical applications. More broadly, this work suggests new strategies for integrating multi-layer information in complex systems and points to the potential of Cube Kernel to reshape network design through efficient multi-scale feature fusion.

## 5 CONCLUSION

In this paper, we propose the Cube Kernel, a lightweight, efficient operator for feature fusion in building segmentation. By restructuring the convolution to permit direct, local gradient coupling across channels, the Cube Kernel restores sensitivity to inter-band interactions. Two key observations stand out from our experiments: **(1) Cube Kernel generalizes across CNN- and Transformer-based backbones, delivering consistent gains without architectural changes; and (2) its efficiency enables stronger accuracy–compute trade-offs.** Concretely, on three widely used datasets, integrating Cube Kernel into CNN- and Transformer-based mainstream backbones yields substantial improvements and achieves state-of-the-art performance with competitive computational cost on the building segmentation task. Moreover, experiments demonstrate that Cube Kernel can be stacked in a manner similar to ResNet blocks, where the resulting networks achieve nearly identical segmentation performance compared to their counterparts, while reducing the parameter count by 42%. Overall, Cube Kernel offers a practical, plug-and-play solution for structure-preserving and compute-efficient cross-channel mixing, and we hope it will serve as a strong default for multi-scale feature fusion in dense prediction. We will release code and models.

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

## USE OF LARGE LANGUAGE MODELS (LLMS).

We used GPT-5 to assist in improving the clarity of writing and formatting of the paper. The model was not involved in the design of experiments, data analysis, or interpretation of results. All responsibility for the content rests with the authors.

## REPRODUCIBILITY STATEMENT.

We are committed to ensuring reproducibility of our results. The anonymized source code and experiment scripts are provided in the supplementary material. The datasets used in this work are publicly available.

## ETHICS STATEMENT

This research relies exclusively on publicly available benchmark datasets (e.g., WHU, Inria Aerial Image Labeling, and Waterloo Building Dataset) that have been widely adopted within the community. These datasets do not contain personally identifiable or sensitive information. We have taken measures to ensure fairness and transparency by providing detailed descriptions of preprocessing steps and releasing anonymized source code.

The intended applications of our method are in domains such as urban planning and disaster response, where accurate building extraction and scene understanding can provide significant societal benefits. Nevertheless, we acknowledge that similar techniques could be misused in surveillance contexts. We therefore stress the importance of responsible and ethical use of the research outcomes presented in this work.

## A  ADDITIONAL RESOURCES

To facilitate reproducibility, we provide anonymized source code and experiment scripts as supplementary material. The repository contains all necessary instructions to reproduce the reported results and ensures compliance with the double-blind review policy while allowing reviewers to verify our implementation.

Anonymous repository link: https://anonymous.4open.science/r/Cube_Kernel_Anonymize-900D/README.md

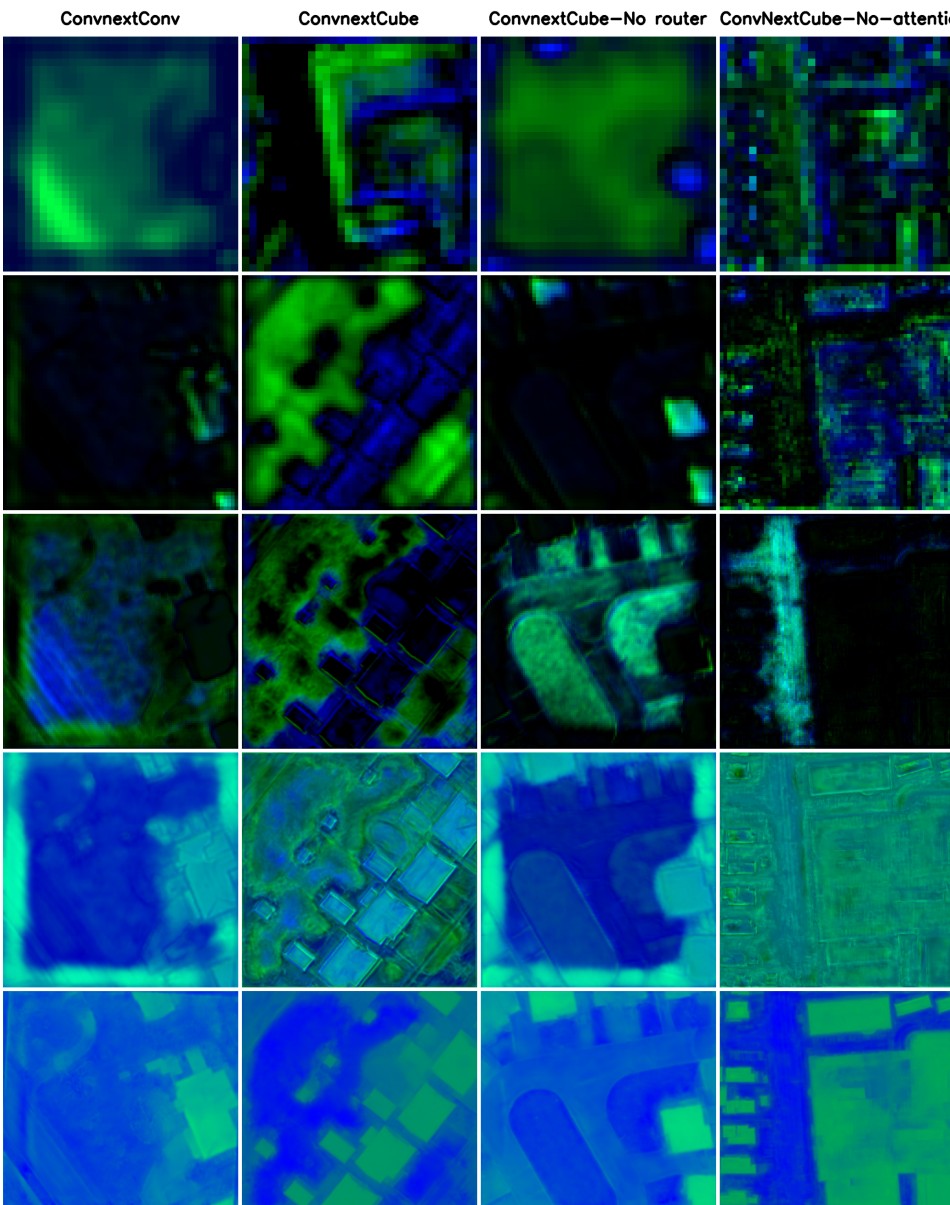

Figure 11: Visualizes the input features for Cube Kernel

## B    ADDITIONAL VISUALIZATIONS

We provide additional qualitative results to further illustrate the effectiveness of Cube Kernel. Specifically, we compare ConvNeXt-Conv, ConvNeXt-Cube, ConvNeXt-No Router, and ConvNeXt-No Attention in Fig. 11.

From the visualizations, we observe that: ConvNeXt-Conv: struggles to capture high-level semantics, and its low-level feature maps contain little useful information. ConvNeXt-Cube (No Router): exhibits degraded performance at deeper layers, as it cannot effectively prioritize task-relevant features and is distracted by irrelevant responses. ConvNeXt-Cube (No Attention): produces noticeably noisier activations, indicating insufficient suppression of redundant information.

These observations collectively demonstrate that the reconstruction, router, and attention components of Cube Kernel are complementary and jointly contribute to more effective extraction of target information.

## C  GRADIENT BEHAVIOR AND CROSS-CHANNEL COUPLING

A key property of Cube Kernel is that it introduces a new backpropagation pathway by reorganizing channel information into shared local neighborhoods. Through grouping and spatial reconstruction, multiple channels co-occur inside the same receptive field, allowing each kernel weight to receive gradients influenced by several channels rather than a single isolated slice.

**Why standard convolution cannot learn cross-channel relationships.**  In a standard convolution, each output channel is computed as

$$Z_i = \sum_{j=1}^{C_{\text{in}}} W_{i,j} * X_j, \tag{3}$$

where cross-channel fusion occurs only through fixed linear summation. The corresponding gradient factorizes into a global term and a channel-isolated local term:

$$\frac{\partial L}{\partial W_{i,j}} = \frac{\partial L}{\partial Z_i} \cdot \underbrace{\frac{\partial Z_i}{\partial W_{i,j}}}_{\text{Local Gradient (Channel-Isolated)}}, \tag{4}$$

where $\frac{\partial Z_i}{\partial W_{i,j}}$ depends only on $X_j$ and is unaffected by all other channels. Thus, no parameter in a standard convolution can receive cross-channel dependent gradients.

**Additive vs. Multiplicative Interactions.**  This limitation becomes clearer by contrasting additive and multiplicative fusion:

$$y = x_1 + x_2, \qquad \frac{\partial y}{\partial x_1} = 1, \quad \frac{\partial y}{\partial x_2} = 1; \tag{5}$$

$$y = x_1 \cdot x_2, \qquad \frac{\partial y}{\partial x_1} = x_2, \quad \frac{\partial y}{\partial x_2} = x_1. \tag{6}$$

Additive fusion produces channel-isolated gradients, whereas multiplicative (or ratio-like) fusion requires cross-channel dependent gradients—something a standard convolution cannot produce due to Eq. 4. This makes proportion-based cues fundamentally inaccessible to a single convolution layer.

**Cube Kernel enables cross-channel gradients.**  After reconstruction, channels within the same group $G$ are spatially interleaved inside a shared $3 \times 3$ receptive field. The gradient for a Cube Kernel weight becomes

$$\frac{\partial L}{\partial W_{\text{Cube},i,j}} = \frac{\partial L}{\partial Z_i} \cdot \underbrace{\frac{\partial Z_i}{\partial W_{\text{Cube},i,j}}}_{\text{Local Gradient (Cross-Channel)}}, \qquad j \in G, \tag{7}$$

where the local term now depends jointly on all channels in $G$. Hence, Cube Kernel provides the missing mechanism that allows cross-channel relationships—especially multiplicative or ratio-like cues—to become directly learnable within one receptive field.

## D  ADDITIONAL EXPERIMENTS ON COCO BENCHMARKS

To assess the generalization of Cube Kernel beyond building extraction, we conduct additional experiments on two large-scale segmentation benchmarks: COCO-Instance80 and COCO-Stuff164K. We consider three representative architectures: (i) a CNN baseline, DeepLabV3+ with a ResNet-50 encoder; (ii) a hierarchical Transformer backbone, Segformer-B2; and (iii) a plain ViT-style decoder,

Table 5: Results on COCO-Instance80 and COCO-Stuff164K using DeepLabV3+ (ResNet-50), SegVit, and Segformer-B2 with and without Cube Kernel.

| Model | mIoU | PA | mPA | FWIoU | Params (M) | FLOPs (G) |
|---|---|---|---|---|---|---|
| DeepLabV3+ + CubeKernel, COCO-80 (21.5h) | 0.3179 | 0.8585 | 0.4021 | 0.7648 | 34.96 | 160.01 |
| DeepLabV3+, COCO-80 (21.5h) | 0.3054 | 0.8548 | 0.3871 | 0.7578 | 39.8 | 187.80 |
| DeepLabV3+ + CubeKernel, COCO-80 (26h) | **0.3419** | **0.8637** | **0.4338** | **0.7742** | 34.96 | 160.01 |
| DeepLabV3+, COCO-80 (26h) | 0.3217 | 0.8583 | 0.4013 | 0.7658 | 39.8 | 187.79 |
| DeepLabV3+ + CubeKernel, COCO-Stuff (39h) | 0.2913 | 0.6184 | 0.4242 | 0.4699 | 34.96 | 160.01 |
| DeepLabV3+, COCO-Stuff (39h) | 0.2877 | 0.6150 | 0.4001 | 0.4609 | 39.8 | 187.79 |
| SegVit + CubeKernel, COCO-Stuff (40h) | **0.3402** | **0.6641** | **0.4470** | **0.5138** | 22.49 | 62.68 |
| SegVit (Zhang et al., 2022), COCO-Stuff (40h) | 0.3277 | 0.6505 | 0.4330 | 0.5074 | 22.68 | 111.97 |
| Segformer-B2 + CubeKernel(add cube), COCO-Stuff (40h) | 0.3082 | 0.6259 | 0.4194 | 0.4775 | 27.54 | 60.22 |
| Segformer-B2 (Xie et al., 2021), COCO-Stuff (40h) | 0.2917 | 0.6109 | 0.4020 | 0.4627 | 27.47 | 59.18 |

Metrics: mIoU, PA, mPA, FWIoU. *Params* = learnable parameters (M). *FLOPs* = MACs for one $3{\times}512{\times}512$ forward pass (G). Training wall-clock time in parentheses. Best in **bold**.

SegVit built on `vit_base_patch16_224`. In DeepLabV3+, we replace only the final $3\times3$ convolution layers in the decoder head with Cube Kernel, keeping all other training settings identical for fair comparison. For Segformer-B2, we insert Cube Kernel at the multi-scale fusion point illustrated in Fig. 7(d) and train the model on COCO-Stuff164K for 40 h under the same data pipeline (without ImageNet pre-training). For SegVit, we replace the depthwise convolutions in the ATM-style decoder with Cube Kernel, again using the same COCO-Stuff164K pipeline and a comparable 40 h training budget.

Using DeepLabV3+ (ResNet-50) as a CNN baseline, Cube Kernel consistently improves mIoU across different training durations on both COCO datasets, while also reducing FLOPs. On COCO-Instance80, Cube Kernel yields gains of +1.25 percentage points mIoU (from 30.54% to 31.79%) at 21.5 h training time and +2.02 percentage points (from 32.17% to 34.19%) at 26 h, while reducing FLOPs from 187.8 G to 160.0 G and parameters from 39.8 M to 35.0 M.

On COCO-Stuff164K, Cube Kernel consistently improves both CNN- and Transformer-based baselines. For DeepLabV3+ (ResNet-50), mIoU increases from 28.77% to 29.13% (+0.36 percentage points), PA from 61.50% to 61.84%, mPA from 40.01% to 42.42%, and FWIoU from 46.09% to 46.99%, while FLOPs are reduced from 187.79 G to 160.01 G under the same 39 h training budget. For the Segformer-B2 backbone on COCO-Stuff164K (training with lr $= 3 \times 10^{-5}$ and inserting Cube Kernel at the fusion point in Fig. 7(d)), Segformer-B2 + CubeKernel, COCO-Stuff (40h) improves mIoU from 29.17% to 30.82% (+1.65 percentage points), PA from 61.09% to 62.59%, mPA from 40.20% to 41.94%, and FWIoU from 46.27% to 47.75%, at only a marginal FLOPs overhead (59.18 G→60.22 G) and almost unchanged parameters (27.47M→27.54M). On a stronger ViT-style decoder, SegVit + CubeKernel, COCO-Stuff (40h) built on `vit_base_patch16_224` with the SegViT_ATM_timm implementation and lr $= 6 \times 10^{-5}$ further improves mIoU from 32.77% to 34.02% (+1.25 percentage points), PA from 65.05% to 66.41%, mPA from 43.30% to 44.70%, and FWIoU from 50.74% to 51.38%, while slightly reducing parameters (22.68M→22.49M) and cutting FLOPs almost in half (111.97 G→62.68 G). For SegVit (ViT-Base/16), we follow the official architecture but replace the original $3\times3$ convolution in the mask decoder with a Cube-Kernel block. Because this convolution operates on full-resolution $512\times512$ feature maps—where both channel count and spatial size make it one of the most parameter- and FLOP-intensive components—substituting it with the more efficient Cube Kernel results in a substantial reduction in FLOPs while preserving the decoder's functionality.

Taken together, Cube Kernel shows modest but consistent improvements under identical training conditions, with low computational cost. These results confirm that Cube Kernel generalizes effectively to diverse scenes and large-vocabulary, multi-class segmentation, and remains beneficial across CNN, hierarchical Transformer, and plain-ViT backbones, rather than being restricted to binary or building-focused datasets.

These results indicate that Cube Kernel remains effective on multi-class and large-scale segmentation, offering stable accuracy gains while reducing computational cost.

## E  ACTIVATION FUNCTION CHOICE: RELU VS. GELU

To understand the role of nonlinear activation in Cube Kernel, we compare ReLU and GELU under identical settings by integrating Cube Kernel into the ConvNeXt architecture. The Channel Router produces mixed-channel responses with diverse magnitudes, including weak texture cues and boundary-sensitive signals. ReLU's hard thresholding (*zeroing all negative responses*) removes low-magnitude activations irreversibly and suppresses gradient propagation, especially problematic after channel reconstruction, where spatially rearranged features rely heavily on retaining fine-grained information.

GELU, in contrast, applies soft, input-dependent gating that preserves the full spectrum of channel-mixed signals while remaining fully differentiable. This avoids the gradient truncation inherent to ReLU and maintains subtle feature variations that are important for Cube Kernel's spatial–channel interactions. This behavior aligns with modern CNN designs: ConvNeXt and ConvNeXtV2 both replace ReLU with GELU in inverted bottleneck structures, validating that GELU is beneficial even outside Transformer-based models.

We integrate Cube Kernel into ConvNeXt and evaluate both activations on the Inria dataset. As shown in Tab. 6, GELU consistently outperforms ReLU across all metrics, improving F1 by +0.90 and IoU by +1.56, highlighting that maintaining fine-grained channel-mixed signals is crucial for Cube Kernel.

Table 6: Ablation on activation functions for Cube Kernel integrated into ConvNeXt.

| Activation | OA (%) | Pre (%) | Rec (%) | F1 (%) | IoU (%) |
|---|---|---|---|---|---|
| ReLU | 96.72 | 90.05 | 88.36 | 89.52 | 81.07 |
| GELU | **97.03** | **92.22** | **88.69** | **90.42** | **82.63** |

These results confirm that GELU provides a more suitable nonlinear transformation for Cube Kernel, enabling better preservation and propagation of mixed-channel information and leading to consistent accuracy improvements.

## F  PARAMETER COMPLEXITY ANALYSIS

To provide a transparent comparison, we report the parameter counts of Cube Kernel and a standard $3 \times 3$ convolution under identical settings: $C_{in} = C_{out} = 96$, kernel size $k = 3$, and grouping factor $G = 4$. All values are computed directly from PyTorch implementations.

**Cube Kernel.**  The proposed Cube Kernel block consists of four lightweight components:

- **Router (1×1 convolution)**: $96 \times 96 = 9{,}216$ parameters.
- **Depthwise 3×3 block**: 96 channels, each with $3 \times 3$ kernels, yielding $96 \times 9 = 216$ parameters.
- **Grouped 3×3 fusion**: grouping factor $G = 4$ gives groups of size 24, leading to 4 groups of $(24 \times 24 \times 3 \times 3)$ parameters, totaling 1,728.
- **Two normalization layers**: 240 learnable scale-and-shift parameters.

The entire Cube Kernel block therefore contains:

$$\text{Total} = 9{,}216 + 216 + 1{,}728 + 240 = \mathbf{13{,}456 \text{ parameters}}.$$

**Standard convolution.**  A conventional $3 \times 3$ convolution with $C_{in} = C_{out} = 96$ contains:

$$96 \times 96 \times 3 \times 3 = \mathbf{36{,}864 \text{ parameters}}.$$

This is $2.74\times$ more than the Cube Kernel under the same configuration, corresponding to a **63.5%** reduction in parameters.

