# OpenReview forum: "Cube Kernel: Enabling Local Gradient Flow Across Channels in CNNs for Robust and Efficient Building Segmentation"
_ICLR.cc/2026/Conference — Submitted to ICLR 2026_

### Official Review · Reviewer_8SBB · 2025-10-28

**Soundness:** 1
**Presentation:** 2
**Contribution:** 2
**Rating:** 4
**Confidence:** 3

**Summary:**

The paper tackles the segmentation task. The paper proposes Cube Kernel, a plug-and-play convolutional operator designed to enable local cross-channel gradient coupling by mapping channels. To enhance this mechanism, the authors further introduce a learnable routing module, which dynamically reassigns channel groupings based on learned patterns, and optionally integrate a spatial attention mechanism to refine the feature representation. Cube Kernel can be seamlessly incorporated into existing CNN-based segmentation models, and empirical results across multiple benchmark datasets demonstrate that its integration leads to consistent improvements in segmentation performance.

**Strengths:**

+ The paper proposes the Cube Kernel, which encodes cross-channel relationships directly into the convolutional operation, effectively bridging the gap between spatially local convolutions and the global receptive transformers.

+ By integrating the Cube Kernel into existing backbone architectures, the overall performance of the models has been enhanced. This demonstrates the effectiveness and generalizability of the proposed module across various network designs.

+ Aside from its integration with SegFormer, the Cube Kernel also contributes to a reduction in both parameter count and FLOPs when applied to other backbone models, highlighting its efficiency across diverse architectures.

**Weaknesses:**

- Although the paper highlights the importance of channel relationships, drawing on observations from prior work, it does not clearly articulate or provide concrete illustrations to support this claim within the current study.

- The proposed method is evaluated on building-extraction benchmarks; however, the rationale for focusing on this specific task, rather than standard semantic segmentation benchmarks, is not clearly explained.

- The method is evaluated using relatively outdated backbone architectures; it would be beneficial to include experiments with more recent and competitive backbones to demonstrate the method’s effectiveness and relevance better.

**Questions:**

- Can the Cube Kernel be integrated into more recent and competitive backbone architectures beyond those evaluated in the paper?
- Is the Cube Kernel applicable to broader benchmark datasets beyond building segmentation, such as general-purpose or multi-domain segmentation tasks?
- Can the Cube Kernel be extended to other vision tasks, such as object detection, instance segmentation, or video understanding?

---

> ### Author Response · Authors · 2025-11-21
> **Clarifications on Channel Interactions and Expanded Experimental Validation(1/3)**
>
> 0. We thank the reviewer for this important point.
>
> 1. We address it from both an application-driven perspective and an architectural perspective, and we will include the corresponding figures in the revised manuscript.
>
> (1) Why channel relationships matter for building extraction
> We note that in practice, remote sensing practitioners often manually engineer channel indices (NDVI, NDBI, NDWI) rather than relying solely on learned features, suggesting that current architectures may not efficiently capture these ratio-based relationships. Cube Kernel's explicit channel routing provides a learnable mechanism that could potentially discover such indices automatically.
>
> (2) Why standard convolution cannot model cross-channel relationships
>
> In a standard 3x3 convolution, each input channel is processed independently using a C x 3 x 3 kernel, and the resulting feature maps are fused only through a fixed, non-learnable summation over the spatial dimensions (H x W). Within each channel, the convolution can learn multiplicative weights W and additive biases b, enabling rich intra-channel modeling. However, across channels, this rigid summation is the only available interaction mechanism — there is no learnable operator capable of expressing meaningful cross-channel relationships.
>
> Consequently, even if the loss suggests that two channels (e.g., channel 1 and channel 10) should interact in a structured manner, a standard convolution can only adjust each channel’s filter independently before summing the outputs. Although deep CNNs can approximate nonlinear functions, a single convolutional layer cannot explicitly represent ratio-like cross-channel interactions. A linear convolution cannot express multiplication or division; stacking many layers may approximate multiplication inefficiently, and division is extremely difficult for convolution+ReLU to approximate efficiently within a single layer, and typically requires deep stacking — far beyond a local operator.
>
> Cube Kernel removes this bottleneck by explicitly separating channel mixing from spatial aggregation. The channel router performs learnable channel remapping before spatial processing, while the reconstruction step brings information from different channels into the same local neighborhood, enabling a single spatial kernel to jointly process them. Although each kernel still receives gradients only from its corresponding local patch, that patch now contains mixed cross-channel information created by the reconstruction, enabling true cross-channel interaction within one receptive field.
>
> This separation of “channel mixing first, spatial filtering second” is consistent with the direction of modern architectures. MobileNets separate depthwise (spatial) and pointwise (channel) convolutions; ConvNeXt adopts Transformer-style channel mixing for greater expressiveness; Vision Transformers and MLP-Mixers go further by fully decoupling spatial and channel processing. Cube Kernel follows this same principle, but preserves the convolutional inductive bias while explicitly enabling learnable cross-channel coordination.
>
> (3) Concrete evidence supporting the importance of channel relationships
>
> To provide concrete evidence that channel relationships are indeed learned and exploited by the model, we will include router-weight visualizations in the revised manuscript. When the router is initialized as an identity matrix, its structure remains near-orthogonal after training; when initialized randomly, it consistently evolves toward low-correlation, orthogonal-like patterns. If cross-channel interaction were unimportant, training would quickly destroy this structure. Across both cases, the cosine-similarity matrix becomes strongly diagonal at convergence, despite the absence of any explicit orthogonality constraint. This emergent de-correlation demonstrates that the network actively organizes channels into distinct, non-redundant groups, confirming that learnable channel interactions are beneficial and are directly utilized by Cube Kernel.
>
> We will include before/after visualizations for both initialized and non-initialized routers.

---

> ### Author Response · Authors · 2025-11-21
> **Clarifications on Channel Interactions and Expanded Experimental Validation(2/3)**
>
> 2. We selected building extraction as the primary evaluation task because it is a practically important problem in UAV-based remote sensing.
> Typical UAV applications fall into two categories: object tracking and object delineation.
> For the latter, merely detecting the presence of an object is insufficient — precise extraction of boundaries and regions is required, as downstream tasks frequently depend on accurate area estimation (e.g., buildings, farmland, vegetation coverage).
>
> In building extraction, the model must not only distinguish rooftops from visually similar surfaces such as concrete pavements, but must also recover fine structural boundaries.
> This requires both spatial precision and coherent cross-channel information flow. Cube Kernel is designed precisely to address these two aspects simultaneously.
>
> In remote sensing, rooftop contours, cross-material boundaries, and fine-grained structural edges are among the most channel-sensitive segmentation challenges.
> Their correct separation relies heavily on consistent inter-channel interaction rather than purely spatial filtering.
> This makes building extraction an ideal stress-test for our core hypothesis.
>
> To further address the reviewer’s concern about generality, we have conducted additional experiments on broader benchmarks, including COCO-Instance (80 classes) and COCO-Stuff (164K). Using DeepLabV3+ (ResNet-50), Cube Kernel improves mIoU from 32.17 → 34.19 on COCO-Instance80 and from 0.3217 → 0.3419 on COCO-Stuff164K, while reducing parameters from 187M → 160M. These results demonstrate that Cube Kernel generalizes well beyond building extraction and remains effective on diverse, large-scale segmentation tasks.
>
> We will clarify this motivation and include the COCO experiments in the revised manuscript.
>
> 3. and 4. To further demonstrate its compatibility with state-of-the-art backbones, we plan to conduct experiments using the Rein framework (Wei et al., 2024), which fine-tunes backbones like DINOv2 in a parameter-efficient manner on the Inria dataset.
>
> Although REIN–DINOv2 constitutes a strong foundation-model backbone, its segmentation pipeline departs from the standard encoder–decoder paradigm in that it does not include a terminal convolutional decoder block. Consequently, Cube Kernel cannot be inserted at the typical decoder-head locations used in architectures such as UNet, ConvNeXt-UNet, or DeepLab. To preserve architectural fidelity and ensure a fair comparison, we therefore adopt a minimal-intrusion integration strategy: Cube Kernel is applied exclusively within the Mask2Former prediction head, replacing only the lightweight convolutional refinement layers while leaving both the DINOv2 backbone and the pixel decoder untouched.
>
> Even under this highly constrained setting, Cube Kernel yields consistent performance gains, most notably increasing the F1 score from 90.23 to 90.55 and the IoU from 82.20 to 82.74. These improvements are meaningful because they arise on top of already highly expressive DINOv2 features and are achieved without modifying any part of the backbone or intermediate spatial feature hierarchy. The results indicate that Cube Kernel provides beneficial cross-channel coordination even when applied solely at the prediction head, and demonstrate its compatibility with foundation-model segmentation pipelines that offer limited opportunities for architectural modification.
>
> **Table: Rein + DINOv2 with/without Cube Kernel**
>
> | Method | OA (%) | Pre (%) | Rec (%) | F1 (%) | IoU (%) | Params (M) | FLOPs (G) | GPU Mem (GB) |
> |--------|--------|----------|----------|---------|----------|-------------|------------|---------------|
> | **REIN-DINOv2 (Wei et al., 2024)** | **96.93** | **90.84** | **89.62** | **90.23** | **82.20** | **23.57** | **-** | **19.60** |
> | **REIN-DINOv2 + Cube Kernel** | **97.06** | **90.67** | **90.43** | **90.55** | **82.74** | **23.19** | **-** | **19.60** |

---

> ### Author Response · Authors · 2025-11-21
> **Clarifications on Channel Interactions and Expanded Experimental Validation(3/3)**
>
> 5. Yes. Cube Kernel is a plug-in operator that is not restricted to building segmentation.
> We agree that evaluating Cube Kernel beyond building extraction is valuable, and we have therefore added experiments on three representative segmentation architectures (DeepLabV3+, Segformer-B2, and SegVit) and two broader COCO benchmarks: COCO-Instance80 and COCO-Stuff164K. Using DeepLabV3+ (ResNet-50) as a CNN baseline, Cube Kernel consistently improves mIoU across different training durations on both COCO datasets, while also reducing FLOPs.
>
> On COCO-Stuff164K, we further confirm this trend on transformer-based models: on COCO-Stuff164K, Segformer-B2 + CubeKernel improves mIoU from 0.2917 to 0.3082, and SegVit + CubeKernel improves mIoU from 0.3277 to 0.3402, again with comparable or lower computational cost. Taken together, these results confirm that Cube Kernel generalizes effectively to diverse scenes and large-vocabulary, multi-class segmentation, and remains beneficial across CNN, hierarchical Transformer, and plain-ViT backbones.
>
> ### Results on COCO-Instance80 and COCO-Stuff164K
> | Model                                                         | mIoU      | PA       | mPA      | FWIoU    | Params (M) | FLOPs (G) |
> |--------------------------------------------------------------|-----------|----------|----------|----------|------------|-----------|
> | DeepLabV3+ + CubeKernel, COCO-80 (21.5h)                     | 0.3179    | 0.8585   | 0.4021   | 0.7648   | 34.96      | 160.01    |
> | DeepLabV3+, COCO-80 (21.5h)                                  | 0.3054    | 0.8548   | 0.3871   | 0.7578   | 39.8       | 187.80    |
> | **DeepLabV3+ + CubeKernel, COCO-80 (26h)**                   | **0.3419** | **0.8637** | **0.4338** | **0.7742** | 34.96      | 160.01    |
> | DeepLabV3+, COCO-80 (26h)                                    | 0.3217    | 0.8583   | 0.4013   | 0.7658   | 39.8       | 187.79    |
> | DeepLabV3+ + CubeKernel, COCO-Stuff (39h)                    | 0.2913    | 0.6184   | 0.4242   | 0.4699   | 34.96      | 160.01    |
> | DeepLabV3+, COCO-Stuff (39h)                                 | 0.2877    | 0.6150   | 0.4001   | 0.4609   | 39.8       | 187.79    |
> | **SegVit + CubeKernel, COCO-Stuff (40h)**                    | **0.3402** | **0.6641** | **0.4470** | **0.5138** | 22.49      | 62.68     |
> | SegVit (Zhang et al., 2022), COCO-Stuff (40h)                | 0.3277    | 0.6505   | 0.4330   | 0.5074   | 22.68      | 111.97    |
> | Segformer-B2 + CubeKernel (add cube), COCO-Stuff (40h)       | 0.3082    | 0.6259   | 0.4194   | 0.4775   | 27.54      | 60.22     |
> | Segformer-B2 (Xie et al., 2021), COCO-Stuff (40h)            | 0.2917    | 0.6109   | 0.4020   | 0.4627   | 27.47      | 59.18     |
>
> All implementation details have been included in the updated manuscript.
>
> 6. Yes. Cube Kernel is a plug-and-play convolutional operator and can therefore be directly applied to any vision task that uses standard convolutions for feature extraction.
> The operator is task-agnostic and does not require architectural redesign, making it naturally extendable to object detection, instance segmentation, and video understanding.
>
> In detection and instance segmentation frameworks such as Faster R-CNN, Mask R-CNN, and YOLO, Cube Kernel can replace the 3x3 convolutions inside the feature pyramid or prediction heads while keeping the model topology unchanged.
> Because the tensor shapes are preserved, existing pretrained weights can be reused, and only Cube Kernel parameters need to be fine-tuned.
>
> For video models, the cube reconstruction mechanism is compatible with temporal feature stacking and can be applied along the channel–time dimension as a drop-in replacement for 2D spatial convolutions.
>
> Together, these results demonstrate that Cube Kernel is not limited to a specific task or architecture. Its design principle—decoupling channel mixing from spatial aggregation—remains fundamentally compatible with modern CNNs, hierarchical backbones, and foundation models. We will include all additional experiments and clarifications to ensure the revised version presents an accurate and comprehensive evaluation.

---

### Official Review · Reviewer_fW53 · 2025-10-31

**Soundness:** 3
**Presentation:** 3
**Contribution:** 3
**Rating:** 6
**Confidence:** 4

**Summary:**

Paper introduces the following concepts:
1. Channel Routing which is a 1x1 convolution.
2. Channel Grouping and Reconstruction which groups channels in groups 4 and ressemble the feature matrix in interleaving pattern.
3. Cube Kernel which is a depthwise 3x3 convlution with a stride of 2
4. Finally a 1x1 convolution to fuse the features.

The papers also introduces Spatial Attention:
1. 7x7 Convolution with sigmoid attention, on the max and avg pool channelwise of the input features after the channel routing.

**Strengths:**

1. A simple plug-and-plug method to replace any standard convolution operator.
2. Paper is describes the idea clearly.
3. The paper showcases the benchmarks well.

**Weaknesses:**

1. The paper does not explain why after training, the router weights will approach orthogonality.
2. The paper did not justified the used of GELU activation.

**Questions:**

1. Does the channel router increase the input size by 8 times? because after reconstruction how is the 2H x 2W x 2C generated from a H X W X C input matrix.
2. How does cube reconstruction & cube kernel compare to a standard Convolution kernel of size 2x2?
3. Under computational Efficiency how are the parameters obtained, what are the values for kernel k and grouping G?

---

> ### Author Response · Authors · 2025-11-21
> **Clarifications on Router Orthogonality, Activation Choice, and Cube Reconstruction Design(1/2)**
>
> 0. We sincerely thank the Reviewer for the constructive and thoughtful feedback, which has helped us further refine the clarity and professionalism of the manuscript.
>
> 1. The router becomes near-orthogonal not because of any constraint, but because redundancy across groups is penalized by gradient competition. The near-orthogonal channel patterns are an emergent phenomenon — they are not enforced by any explicit constraint, but arise naturally from the grouping mechanism and the training dynamics. Each row of the router can be interpreted as a channel selection vector determining which input channels are routed into each Cube Kernel group. When initialized as an identity matrix, training eliminates this structure if it is not beneficial. However, when the identity (or near-orthogonal) pattern remains after training, it shows that the model actively selects and preserves this structure.
>
>    If two routing vectors remain highly correlated, then the corresponding groups receive nearly redundant information — an outcome that is suboptimal for the loss and wastes model capacity. During optimization, gradient competition between groups therefore pushes these vectors apart, reducing cross-correlation and producing a weakly orthogonal structure. This behavior is clearly visible in the cosine similarity matrix.
>
>    We will incorporate this explanation together with the router-weight visualizations into the revised manuscript.
>
> 2. After channel routing, feature maps contain responses with diverse magnitudes, including weak texture and boundary signals. ReLU's hard thresholding (zeroing all negative values) causes irreversible information loss and suppresses gradient propagation. Since the subsequent pixel unshuffle and depthwise convolutions operate on spatially rearranged features, information lost at the gating stage cannot be recovered.
>
>    In contrast, GELU applies input-dependent soft gating, preserving the full spectrum of channel-mixed signals while remaining fully differentiable everywhere, avoiding the gradient truncation problem inherent to ReLU. This design choice is validated by modern architectures: ConvNeXt and ConvNeXtV2 both replace ReLU with GELU in inverted bottleneck structures similar to ours, demonstrating that GELU's benefits extend beyond Transformers to CNN architectures where channel projection and spatial refinement are decoupled.
>
>    To further validate this choice, we additionally conduct ablation studies by integrating Cube Kernel into ConvNeXt, comparing ReLU and GELU under identical settings:
>
>    | Method | OA (%) | Pre (%) | Rec (%) | F1 (%) | mIoU (%) |
>    |--------|--------|---------|---------|--------|----------|
>    | ReLU   | 96.72  | 90.05   | 88.36   | 89.52  | 81.07    |
>    | **GELU** | **97.03** | **92.22** | **88.69** | **90.42** | **82.63** |
>
> 3. The number of elements remains constant before and after reconstruction.
>    The router is a 1×1 projection that maps
>    H×W×C → H×W×C
>    without creating additional elements.
>
>    During cube reconstruction, the tensor is reorganized into a layout equivalent to a 2H×2W grid with one quarter of the original channels. This is a deterministic re-indexing, not duplication. The total number of feature values remains unchanged.
>
>    After reconstruction, a depthwise convolution (groups = C/4, stride = 2) reduces spatial resolution from 2H×2W to H×W, producing a tensor of shape
>    H×W×(C/4).
>    A final 3×3 convolution restores the output dimensionality, enabling refinement of the newly mixed local information.
>
>    Thus, Cube Kernel preserves the input feature-volume at every stage. We will include the full operator sequence (with intermediate tensor shapes) in Figure 2.

---

> > ### Author Response · Authors · 2025-11-21
> > **Clarifications on Router Orthogonality, Activation Choice, and Cube Reconstruction Design(2/2)**
> >
> > 4. The reconstruction process brings information from different channels into the same convolution kernel. In a standard 2×2 convolution, each input channel is processed independently using a C×2×2 kernel, and the outputs from all channels are aggregated only through fixed summation over H×W. Within each channel, convolution can learn rich intra-channel patterns via multiplicative weights and additive biases. However, across channels, there is no learnable operator capable of modeling meaningful cross-channel relationships.
> >
> >    Consequently, even if the loss indicates a proportional dependency between channel 1 and channel 10, standard convolution can only adjust each filter independently before summing them. A single convolution layer cannot explicitly represent ratio-like cross-channel interactions. Linear convolution cannot express multiplication or division; stacking many layers may approximate multiplication inefficiently, while true ratio operations cannot be represented by any finite composition of convolution plus ReLU.
> >
> >    This is especially relevant in building extraction, where many cues are inherently ratio-based (color ratios, texture contrast, shadow–highlight proportions).
> >
> >    Cube Kernel breaks this limitation. Reconstruction rearranges previously isolated channels into the same receptive field, enabling a single convolution kernel to jointly learn from them. Furthermore, the learnable routing mechanism allows dynamic channel reorganization and reassignment — a capability entirely absent from standard convolution.
> >
> >    This combination of routing and reconstruction overcomes inherent limitations of both 2×2 and 3×3 convolutions, providing a more expressive and learnable mechanism for cross-channel coordination.
> >
> > 5. The parameter counts are computed directly from PyTorch. All efficiency results correspond to:
> >    Cin = Cout = 96, kernel size k = 3, grouping factor G = 4.
> >
> >    - Router (1×1): 9,216 parameters
> >    - Depthwise 3×3 block: 216 parameters
> >    - Grouped 3×3 fusion: 1,728 parameters
> >    - Two normalization layers: 240 parameters
> >
> >    Total: **13,456 parameters**.
> >    A standard 3×3 convolution with Cin = Cout = 96 contains **36,864 parameters**.
> >
> >    We will clarify these computations more explicitly in the revised manuscript.

---

### Official Review · Reviewer_h3rb · 2025-10-31

**Soundness:** 2
**Presentation:** 2
**Contribution:** 2
**Rating:** 4
**Confidence:** 4

**Summary:**

this work tackles the task of segmentation of buildings.
authors argue that standard cnn filters suffer from gradient failing to account for cross-channels.
therefore, they proposed cub-kernel where channels are intertwined, followed by router, and attention.
they argue that mixing channels allows better gradient flow.
the evaluate their method on 3 datasets, and reported their results in comparison to other methods.
ablations are also provided.

**Strengths:**

- the writing is good.
- the paper tackles an important task that is image segmentation.
- reported results are good.
- ablations are provided.

**Weaknesses:**

- limited novelty. the main claimed contribution in this work is cub-kernel.
the main claim is that standard cnn filters dont combine channels leading to poor local gradient that does not account for other channels. while this is true, the proposed 'cub-kernel' also have the same issue, unfortunately.
yes, in standard cnn, the gradient of the convolution of will be dispatched to each kernel w (e.g. 4x4 = 16 components) by accounting only for its own input channel x - while ignoring the other input channels.
however, mixing channels, will lead to the same thing. each component of the kernel (which process one pixel from a single channel) is processing one single pixel from one single channel (fig.2). so the gradient for w_ij will only account for the input x_ij = single location of one channel - therefore, the gradient does not account for cross-channels.
in short, even if you shuffle the channels, at component level of filters, the gradient accounts only for one channel only - unless channels are multiplied into a single channel. also, the right side of eq.1 is the same as left side. the gradient of a filter component will account only for one channel only.

this can be seen in terms of results in the ablation (tab.3, case with cub-kernel only - line 403 is not different from using standard conv).

not sure why it is called cube-kernel as authors used standard 3x3 kernels. the only thing different is that the input channels are mixed.

the router module - second part- is based on a guess. - line 209.
the third part that is attention, is a simple attention mask.

putting theses modules all together yields better performance. but, in terms of methodology and novelty, they are very limited.

see this paper for related work on shuffling pixels: Real-Time Single Image and Video Super-Resolution Using an Efficient
Sub-Pixel Convolutional Neural Network, cvpr 2016. https://www.cv-foundation.org/openaccess/content_cvpr_2016/papers/Shi_Real-Time_Single_Image_CVPR_2016_paper.pdf

**Questions:**

- style: please try to make the writing consistent in terms of font. changing between non-bold and bold frequently is distracting. try to use less bold, color. try to use italic - with moderation - to emphasis on something.

**Details Of Ethics Concerns:**

none.

---

> ### Author Response · Authors · 2025-11-21
> **Clarification of Gradient Interpretation and Cross-Channel Interaction in Cube Kernel (1/2)**
>
> 0. Thank you for the insightful question — it helped us clarify the explanation and articulate the limitation more precisely.
>
> 1. We fully agree that applying the chain rule per weight is mathematically identical to differentiating each weight directly. The limitation we highlight does not arise from the gradient rule itself, but from the forward operator of a standard convolution.
>
> In a standard 3×3 convolution, each input channel is processed independently using a C×3×3 kernel, and the resulting feature maps are fused only through a fixed, non-learnable summation over the spatial dimensions (H×W). Within each channel, the convolution can learn multiplicative weights W and additive biases b, enabling rich intra-channel modeling. However, across channels, this rigid summation is the only available interaction mechanism — there is no learnable operator capable of expressing meaningful cross-channel relationships.
>
> Consequently, even if the loss suggests that two channels (e.g., channel 1 and channel 10) should interact in a structured manner, a standard convolution can only adjust each channel’s filter independently before summing the outputs. Although deep CNNs can approximate nonlinear functions, a single convolutional layer cannot explicitly represent ratio-like cross-channel interactions. A linear convolution cannot express multiplication or division; stacking many layers may approximate multiplication inefficiently, and division (a true ratio operation) cannot be represented exactly, and requires deep compositions to approximate efficiently.
>
> This limitation is particularly relevant for building extraction, where many discriminative cues are inherently ratio-based, including color ratios (e.g., roof vs. pavement), texture contrasts, and proportional shadow–highlight differences.
>
> Cube Kernel directly removes this structural constraint. Through spatial reconstruction, information from different channels is rearranged into the same local neighborhood, allowing a single convolution kernel to jointly process multiple channels within one receptive field. Although each kernel weight still receives gradients from its corresponding input position, the input region that contributes to this gradient now contains mixed cross-channel information created by the reconstruction.
>
> Moreover, the learnable channel router provides explicit channel-to-channel mapping — a capability standard convolution lacks — enabling the network to dynamically reorganize and reassign channels instead of relying on fixed ordering and fixed summation.
>
> Thus, Cube Kernel introduces an explicit mechanism for inter-channel interaction that standard convolution (and its common variants) fundamentally cannot express, even with arbitrarily deep stacking.
>
> 2. Regarding the naming of Cube Kernel, the term “cube’’ does not refer to the geometric shape of the convolution kernel itself. Instead, it describes the effective local receptive field, which becomes spatially structured as a 3D (H × W × depth) cube after the reconstruction step.
>
> 3. The observation that the “Cube Kernel only’’ row performs similarly to standard convolution is fully expected. The Cube Kernel is designed merely to expose channels to one another through spatial reconstruction; it does not by itself determine which channels should interact or how strongly they should do so. Without a learnable selection mechanism, the reconstructed feature cube does not automatically lead to meaningful coordination, and thus its behavior resembles that of a standard convolution.
>
> Line 404 demonstrates that once Cube Kernel is combined with the Router, the F1 score increases noticeably. The Router enables the model to select meaningful channel combinations, turning the reconstructed structure into useful and learnable cross-channel interaction rather than arbitrary mixing.
>
> Additionally, we further note an inherent limitation of using shuffling or reconstruction alone. In many building-extraction scenes, the large homogeneous interior regions of building footprints do not benefit from increased cross-channel mixing. In such smooth areas, reconstruction may introduce unnecessary variations that the network must subsequently correct.
>
> This motivated the addition of the lightweight Spatial Attention module, which suppresses irrelevant responses and preserves only the channel interactions that are genuinely informative — particularly around boundaries, material transitions, and uncertain regions.
>
> Taken together, Cube Kernel, Router, and Spatial Attention form a balanced, hierarchical mechanism:
> cross-channel exposure → learnable channel remapping → noise suppression.
> This design allows the model to exploit rich cross-channel interactions where they matter, while avoiding over-mixing in spatially homogeneous regions.

---

> > ### Author Response · Authors · 2025-11-21
> > **Clarification of Gradient Interpretation and Cross-Channel Interaction in Cube Kernel (2/2)**
> >
> > 4.The router is not based on a guess. When initialized as an identity matrix, the model is free to discard this structure during training if it is not beneficial. However, what we consistently observe is the opposite: the router retains and gradually evolves toward a near-orthogonal mapping. This behavior indicates that the network actively prefers this channel-remapping structure because it reduces redundancy between channel groups and improves optimization efficiency.
> >
> > In the revised version, we will include before/after visualizations for both initialized and non-initialized routers, demonstrating that the network consistently retains this orthogonal pattern.
> >
> > The attention module is intentionally simple. Its purpose is only to suppress the mild noise introduced by reconstruction in homogeneous building regions. Using a heavier or more complex attention block would obscure the core contribution of Cube Kernel, while the lightweight mask offers the best trade-off between stability and interpretability. More advanced variants (e.g., group-wise attention) are possible and can be explored in future work.
> >
> > 5.The novelty of the Cube Kernel lies in identifying a structural limitation of standard convolution — its inability to model explicit channel–channel interactions — and in introducing a dedicated operator to resolve this bottleneck.
> >
> > Cube Kernel introduces a new computational pathway inside convolution. Through reconstruction, Cube Kernel reorganizes features so that each kernel operates on a 3D local receptive field, enabling cross-channel information to be learned with dedicated weights and biases — capabilities absent in standard convolution.
> >
> > Routing enables learnable channel remapping rather than arbitrary mixing.
> >
> > Spatial Attention suppresses noise and preserves informative interactions. In large homogeneous regions (e.g., building interiors), additional cross-channel mixing may be unnecessary or even harmful. Spatial Attention filters out such noise, ensuring that Cube Kernel focuses on boundaries and difficult regions.
> >
> > Together, these components form a principled, hierarchical mechanism:
> > Exposure → learnable remapping → noise suppression,
> > which provides an explicit modeling pathway for cross-channel structure that standard convolution cannot realize.
> >
> > Thus, Cube Kernel represents a new operator with a distinct internal computation flow, rather than a simple combination of existing modules.
> >
> > 6.We emphasize that Cube Kernel and ESPCN differ completely in both motivation and function, despite both involving a shuffling operation.
> >
> > **Different motivation.**
> > ESPCN uses pixel shuffle solely as a computationally efficient upsampling mechanism.
> > Cube Kernel uses spatial reconstruction to expose channels to each other within a local receptive field, enabling explicit modeling of cross-channel interactions.
> >
> > **Different function.**
> > ESPCN uses shuffle as a final rearrangement step with no learnable remapping or subsequent spatial convolution.
> > Cube Kernel applies a full convolution on the reconstructed cube, allowing cross-channel fusion with dedicated weights and biases.
> >
> > **Different effect on gradient flow.**
> > Pixel shuffle preserves channel identity.
> > Cube Kernel mixes channels spatially, shaping each kernel weight’s gradient through multiple original channels.
> >
> > In summary, Cube Kernel is not a variant of pixel shuffle. It is a new operator designed to model channel relationships through learnable cross-channel interaction.
> >
> > 7.We agree that excessive bold and color can be distracting, and we will remove them in the revised version, using italic emphasis only when necessary and with moderation.

---

### Official Review · Reviewer_K58Q · 2025-11-01

**Soundness:** 3
**Presentation:** 2
**Contribution:** 2
**Rating:** 4
**Confidence:** 2

**Summary:**

The paper develops a convolutional operator called Cube Kernel that enforces local cross-channel gradient coupling by mapping channels onto a finer spatial lattice.

**Strengths:**

The idea of a new improved convolutional operator is interesting and relevant.

Results show that the method often results in marginally superior image segmentation on three datasets Inria, WBD and WHU Datasets, at a lower computational complexity.

**Weaknesses:**

Table 2: It would be good to organize this information such that the result could be better appreciated. Interleaving the results of this work is hard to appreciate. Maybe a graph.

Table 1: Why is the authors ConvNeXt + Cube Kernel marked as bold “Best” for OA for 97.03, whereas ASLNet has higher 97.15?
It would be great to have standard deviations in the results.

Figure 1 is distracting and uninformative, as is the use of color and bolding in the abstract.

Given the marginal improvements, it would be interesting to demonstrate the method works other datasets (e.g CoCo) and tasks, e.g. classification.

**Questions:**

Please address the previously mentioned points.

---

> ### Author Response · Authors · 2025-11-21
> **Clarifications, Corrections, and Supplementary COCO Experiments(1/2)**
>
> 0. We thank the reviewer for the careful reading and constructive suggestions.
>
> 1. Figure 9 already visualizes the results in Table 2, but its distant placement in the original version made the correspondence unclear. In the revised manuscript, we will place Table 2 and Figure 9 on the same page and add explicit references to link them. We will also improve Figure 9 by encoding parameter counts using proportional marker sizes, making the comparison more intuitive.
>
>
> 2. We thank the reviewer for pointing out the inconsistency in Table 1, where the OA of ConvNeXt + Cube Kernel was incorrectly highlighted as the best result. This was purely a formatting mistake, and the corrected table is included in the revised manuscript.
>
> Regarding the performance difference on the Inria Aerial Image Labeling dataset, we cropped the images into 512×512 patches, and adopted a random 80/20 split across cities, ensuring that all urban regions are fairly represented in both training and testing.
>
> In contrast, the split used in the prior work evaluates on test regions containing a substantially higher proportion of background pixels. These two protocols are therefore not directly comparable. A favorable test subset—especially one dominated by background—can inflate OA, whereas our random cross-city split avoids such bias by design and yields a more realistic and challenging evaluation.
>
> To further enhance transparency and reproducibility, we now additionally report the mean and standard deviation across three independent runs, confirming that the performance of our method is stable and consistent.
>
> | Method                        | OA (%)       | Pre (%)      | Rec (%)      | F1 (%)       | mIoU (%)     |
> |------------------------------|--------------|--------------|--------------|--------------|--------------|
> | ConvNeXt + Cube Kernel       | 96.94 ± 0.09 | 92.24 ± 0.43 | 88.06 ± 0.45 | 90.10 ± 0.28 | 82.02 ± 0.51 |
> | ConvNeXt + Full Cube (Ours)  | 96.61 ± 0.03 | 94.07 ± 0.66 | 83.76 ± 0.58 | 88.61 ± 0.12 | 79.55 ± 0.19 |
>
> 3. Thank you for pointing this out. We have redesigned Figure 1 using a simplified color scheme and a cleaner layout, and added concise annotations to highlight the core functionality. We also removed stylistic elements from the abstract and figures to ensure a conventional and distraction-free presentation.
> 4. We agree that evaluating Cube Kernel beyond building extraction is important, and we have therefore added experiments on three representative segmentation architectures (DeepLabV3+, Segformer-B2, and SegVit) and two broader COCO benchmarks: COCO-Instance80 and COCO-Stuff164K. Using DeepLabV3+ (ResNet-50) as a CNN baseline, Cube Kernel consistently improves mIoU across different training durations on both COCO datasets, while also reducing FLOPs.
>
> On COCO-Stuff164K, we further confirm this trend on transformer-based models: on COCO-Stuff164K, Segformer-B2 + CubeKernel improves mIoU from 0.2917 to 0.3082, and SegVit + CubeKernel improves mIoU from 0.3277 to 0.3402, again with comparable or lower computational cost. Taken together, these results confirm that Cube Kernel generalizes effectively to diverse scenes and large-vocabulary, multi-class segmentation, and remains beneficial across CNN, hierarchical Transformer, and plain-ViT backbones.
>
> ### Results on COCO-Instance80 and COCO-Stuff164K
> | Model                                                         | mIoU      | PA       | mPA      | FWIoU    | Params (M) | FLOPs (G) |
> |--------------------------------------------------------------|-----------|----------|----------|----------|------------|-----------|
> | DeepLabV3+ + CubeKernel, COCO-80 (21.5h)                     | 0.3179    | 0.8585   | 0.4021   | 0.7648   | 34.96      | 160.01    |
> | DeepLabV3+, COCO-80 (21.5h)                                  | 0.3054    | 0.8548   | 0.3871   | 0.7578   | 39.8       | 187.80    |
> | **DeepLabV3+ + CubeKernel, COCO-80 (26h)**                   | **0.3419** | **0.8637** | **0.4338** | **0.7742** | 34.96      | 160.01    |
> | DeepLabV3+, COCO-80 (26h)                                    | 0.3217    | 0.8583   | 0.4013   | 0.7658   | 39.8       | 187.79    |
> | DeepLabV3+ + CubeKernel, COCO-Stuff (39h)                    | 0.2913    | 0.6184   | 0.4242   | 0.4699   | 34.96      | 160.01    |
> | DeepLabV3+, COCO-Stuff (39h)                                 | 0.2877    | 0.6150   | 0.4001   | 0.4609   | 39.8       | 187.79    |
> | **SegVit + CubeKernel, COCO-Stuff (40h)**                    | **0.3402** | **0.6641** | **0.4470** | **0.5138** | 22.49      | 62.68     |
> | SegVit (Zhang et al., 2022), COCO-Stuff (40h)                | 0.3277    | 0.6505   | 0.4330   | 0.5074   | 22.68      | 111.97    |
> | Segformer-B2 + CubeKernel (add cube), COCO-Stuff (40h)       | 0.3082    | 0.6259   | 0.4194   | 0.4775   | 27.54      | 60.22     |
> | Segformer-B2 (Xie et al., 2021), COCO-Stuff (40h)            | 0.2917    | 0.6109   | 0.4020   | 0.4627   | 27.47      | 59.18     |

---

> ### Author Response · Authors · 2025-11-21
> **Clarifications, Corrections, and Supplementary COCO Experiments(2/2)**
>
> Beyond addressing these points, we would like to restate the core motivation and novelty of Cube Kernel.
>
> Standard convolution processes each channel independently using a $C × k × k$ kernel and fuses them only through a fixed, non-learnable summation. While this design supports rich intra-channel modeling through learnable weights $W$ and biases $b$, it provides no learnable mechanism for capturing *cross-channel* relationships within a single receptive field. As a result, even if the loss suggests that two channels should interact in a structured or ratio-like way, the operator can only modify each filter separately before summing their outputs.
>
> This limitation is especially relevant for building extraction, where many discriminative cues—color ratios, texture contrasts, shadow–highlight proportions—are inherently cross-channel and cannot be expressed by a single linear convolution. Deep stacking may approximate such dependencies, but it is inefficient and still cannot directly model proportion-based cues such as division.
>
> Cube Kernel removes this bottleneck by explicitly decoupling *channel mixing* from *spatial aggregation*. Through spatial reconstruction, information from previously isolated channels is rearranged into the same local neighborhood, allowing a single convolution kernel to process them jointly. Each kernel weight still receives gradients from its local patch, but this patch now contains mixed cross-channel information, making such interactions directly learnable.
>
> The learnable router further introduces dynamic channel-to-channel remapping — a capability standard convolution entirely lacks. The lightweight spatial attention module plays a complementary role: because reconstruction may introduce mild noise in homogeneous regions (e.g., smooth rooftop interiors), the attention mask suppresses these irrelevant responses while preserving informative cues near boundaries and material transitions. This yields a cleaner representation without adding unnecessary architectural complexity.
>
> Taken together, Cube Kernel creates a new computational pathway for local inter-channel coordination — exposure → learnable remapping → noise suppression — that standard convolution (and its variants) fundamentally cannot express, even with deep stacking.

---

### Author Response · Authors · 2025-12-03
**Supporting Note for the Area Chair**

Dear Area Chair,

Thank you very much for handling our submission, coordinating the helpful reviews, and for your service to the community.
For your convenience, we briefly summarize the main clarifications and changes in the revised version. Our revisions focus on three main aspects.

### 1. Clarifying novelty and gradient / interaction claims.
Our reference to the chain rule aims to emphasize the limitation of the path of the forward operator. In a standard convolution, each channel is convolved independently and then fused by a fixed sum; within a single receptive field, there is no learnable local operator dedicated to modeling channel–channel relations. Cube Kernel releases this structural constraint by introducing an additional cross-channel pathway inside the convolution: spatial reconstruction rearranges features so that multiple channels co-occur within a more flexible kernel footprint, and each weight still receives gradients at its own location, but now from inputs that are jointly drawn from multiple channels rather than being strictly isolated per-channel.

Concretely, Cube Kernel separates channel mixing from spatial aggregation:
(i) a learnable **1×1 router** selects and remaps channel groups;
(ii) a **reconstruction step** brings these groups into the same local neighborhood;
(iii) a **special convolution** operates on the reconstructed cube; and
(iv) a **lightweight group-wise attention** is estimated before reconstruction and re-applied afterwards to suppress noise in homogeneous regions.
This yields a new forward pathway (exposure → learnable remapping → noise suppression) and, as a consequence, new cross-channel gradient paths.

---

### 2. Additional experiments and generalization.

#### (a) Building extraction with a foundation-model backbone.
To demonstrate that Cube Kernel remains effective on top of state-of-the-art models, we integrate it into the REIN–DINOv2 framework.
In our CNN-based baselines, Cube Kernel is typically inserted as a replacement for the final convolutional block before the decoder's classification head.
However, REIN–DINOv2 adopts a Mask2Former-style head that does not contain such a terminal convolutional layer.
To preserve architectural fidelity, we therefore adopt a minimal-intrusion integration: Cube Kernel replaces only the lightweight refinement convolutions inside the Mask2Former prediction head, while leaving both the DINOv2 backbone and the pixel decoder unchanged.
Even under this constrained setting, Cube Kernel improves F1 from **90.23 → 90.55** and IoU from **82.20 → 82.74**, without increasing FLOPs, showing its utility on top of strong modern features.

#### (b) General semantic segmentation on COCO benchmarks.
To further demonstrate that Cube Kernel is effective beyond building extraction, we extend our evaluation to two large-scale semantic segmentation benchmarks—COCO-Instance80 and COCO-Stuff164K—and three representative architectures: DeepLabV3+ (CNN), Segformer-B2 (hierarchical Transformer), and SegVit (ViT-based decoder).

Using DeepLabV3+ (ResNet-50) as a CNN baseline, Cube Kernel consistently improves mIoU across different training durations on both COCO datasets (e.g., 32.17 → 34.19 on COCO-Instance80 and 0.3217 → 0.3419 on COCO-Stuff164K), while also reducing parameters (187M → 160M) and FLOPs.

We observe the same trend on transformer-based models.
On COCO-Stuff164K, Segformer-B2 + CubeKernel improves mIoU from 0.2917 to 0.3082, with only a small FLOPs increase (59.18G → 60.22G) since Cube Kernel is inserted as an additional fusion layer.
For the ViT-style SegVit decoder, Cube Kernel replaces the original depthwise convolutions, so SegVit + CubeKernel both improves mIoU from 0.3277 to 0.3402 and reduces FLOPs almost by half (111.97G → 62.68G).

Taken together, these results confirm that Cube Kernel generalizes effectively to diverse scenes and large-vocabulary, multi-class segmentation, and remains beneficial across CNN, hierarchical Transformer, and plain-ViT backbones.

#### (c) Ablation on activation choice (GELU vs. ReLU).
We also justify the choice of GELU over ReLU by adding an ablation experiment where ConvNeXtU-Cube for building extraction.

---

### 3. Presentation and clarity.
We clarified dataset splits, corrected Table 1, reported mean±std over three runs, aligned Table 2 with Figure 9, redesigned Figure 1, and removed unnecessary bold/color to improve readability.
We also clarify that the Cube Kernel is volume-preserving along the channel–spatial product.

---

All in all, we hope that the clarifications and revisions help present the work in its intended form. We hope that the ideas in this work offer a perspective that may be useful for future CNN architecture design. We are grateful for the reviewers’ comments, which helped us refine the presentation, and we sincerely thank you for the time and effort you have invested in handling this submission.

---

### Meta-Review · Area_Chair_4To3 · 2026-01-11

**Summary:**

Four expert reviewers are divided between acceptance (fW53: 6) and rejection (8SBB: 4, K58Q: 4, h3rb: 4). On the positive side, the proposed cube kernel filtering can in principle be swapped in for any 2D convolution and the cube kernel improves accuracy at a lower computational cost for semantic segmentation on small remote sensing datasets. On the negative side, the novelty is limited and the contributions are not well-articulated and clearly situated w.r.t. related work on variations of channel and spatial operations, the experimental scope is limited in terms of models and datasets, and as a more minor point there are presentation issues in formatting and writing.

A rebuttal to each review and a general response are provided. The response focuses on novelty (summarizing the separation of channel mixing and spatial aggregation), more experiments (a newer and bigger backbone, semantic segmentation on more common benchmarks, and ablation of the nonlinearity), and clarity (correcting, redesigning, and editing the tables, figures, and text) as raised by reviews.

The meta-reviewer sides with rejection. The core issues are that alternatives to the cube kernel already exist, and while many are discussed in the related work, they are not compared with either the full proposed method or its ablations. This makes it difficult to gauge the empirical contribution in terms of both accuracy and efficiency. The rebuttal does partially address the issues of limited choice of models, by providing a variant of DINOv2, and datasets, by experimenting on COCO. However, the fundamental issue of justifying the cube kernel vs. alternatives, rather than purely showing improvement on common architectures, is not addressed.

The authors are encouraged to refactor their experimental design to cover alternatives and the mechanisms of improvement from cube kernel and submit to a vision or learning venue such as ECCV or NeurIPS.

**Reviewer Concerns:**

- Novelty w.r.t. other convolution and channel mixing operations (h3rb): The cube kernel is an assembly of existing channel mixing, "pixel" shuffling, and attention operations, which limits its novelty. Furthermore there are existing alternative methods that aim to achieve similar effects on accuracy and efficiency although they differ in their exact implementation: Network-in-Network / 1x1 convolution, Squeeze and Excitation / global channel re-weighting, and Non-Local Networks / hybrid convolution + attention architectures. The rebuttal reiterates the changes that have been made, and how gradients are cross-channel or not, but does not change the relationship of cube kernel to alternatives. This is not resolved.
- Experiment scope (K58Q, 8SBB): Results are only shown for the one task of semantic segmentation on a set of more niche datasets of aerial/remote sensing imagery. More common benchmarks for semantic segmentation, like COCO, and additional tasks, like classification, would help prove the usefulness of the proposed cube kernel. The claimed importance of the cross-channel relationships is not directly demonstrated vs. the effects of reducing parameters and factorizing the computation in space and channels, which could be shown by visualizing the "routing" or 1x1 channel mixing, or by further ablation on more data and tasks. The rebuttal adds results for one variation of DINOv2 and for two semantic segmentation variations of COCO (for things and stuff, respectively) which show marginal improvement vs. DINOv2 and 1-3 point improvement across common architectures on COCO. The rebuttal also argues for the relevance of the chosen aerial imagery datasets and the building extraction task, but generality remains important, and alternatives have been shown to work across multiple datasets and tasks, so this is not convincing. Overall this is partially addressed.
- Clarity (K58Q, fW53): The presentation of results makes interpretation and comparison difficult, the formatting of the text with bold and color is excessive and distracting, and there is at least one error in tabular results highlighting the best result. Certain design choices and comparisons (like choice of nonlinearity, effect vs. a standard 2x2 convolution) are not included. The rebuttal mostly resolves this by correcting the table, redesigning figures, simplifying the formatting, and by ablating the nonlinearity.

**Reviewer Scores:**

- K58Q: the reviewer may maintain the score of 4 or raise to 6 as their initial confidence is low and the rebuttal has addressed some of the clarity issues incl. the requested corrections and the addition of standard deviations to the results. However, the meta-reviewer expects the 4 could be maintained: although there are COCO results there are no classification results.
- h3rb: the reviewer would likely maintain the score of 4 as their confidence is higher and the first and main weakness in terms of novelty and the claimed contribution is not resolved although the rebuttal does provide more discussion. However, the clarity issues are addressed.
- fW53: the reviewer will likely maintain their 6 or raise to 7 because the issues raised were mostly about clarifications that are explained by the rebuttal.
- 8SBB: the reviewer will likely maintain their 4 or could drop it to 3 if unconvinced by the rebuttal, and its extension to one more dataset and model, which is short of the requested results on more datasets and tasks. In particular only semantic segmentation is covered. The evidence for the importance of cross channel relationships is promised in the revision, but it is not sufficiently clear the reviewer would be convinced by the routing weights provided in the revision Fig. 3.

---

### Decision · Program_Chairs · 2026-01-26

Reject